# communications
# engineering

# Haptics based multi-level collaborative steering control for automated driving

Tomohiro Nakade [1,2✉], Robert Fuchs[2], Hannes Bleuler[3] & Jürg Schiffmann[1]

Increasing the capability of automated driving vehicles is motivated by environmental, productivity, and traffic safety benefits. But over-reliance on the automation system is known to cause accidents. The role of the driver cannot be underestimated as it will ultimately be the most relevant aspect for trust building and social acceptance of this technology. Here we introduce a driver-oriented automation strategy to achieve collaborative steering. Our approach relies on three major functionalities: interaction, arbitration, and inclusion. The proposed control strategy is grounded in the concept of shared control enabling driver intervention over the automation without deactivation. Well-defined physical human-robot interaction types are made available with the arbitration strategy. The automated driving trajectory is adapted to include the driver intent into the tactical level of trajectory planning. This enables driver initiated rerouting and consistent coordination of all vehicle actuators. In this way, automated vehicles, which rely on sight only, are augmented with the incorporation of the driver intent. The driver is neither replaced by nor excluded from the automation, rather their role remains active to the benefit of trust building and driving safety.

[1] Laboratory for Applied Mechanical Design (LAMD), Ecole Polytechnique Fédérale de Lausanne (EPFL), 2002 Neuchatel, Switzerland. [2] Systems Innovation R&D Department, JTEKT Corporation, Kashihara 634-8555, Japan. [3] School of Engineering, Ecole Polytechnique Fédérale de Lausanne (EPFL), 1015 Lausanne, Switzerland. ✉email: tomohiro_nakade@jtekt.co.jp

Advanced driver assistance systems (ADAS) used in partial automation are intended to reduce the driver workload without causing disengagement. Level-2 automation splits the responsibility of the real-time operational and tactical functions to operate a vehicle safely in on-road traffic[1], where the driver is responsible for the 'object and event detection and response (OEDR)', while the vehicle operates the sustained lateral and longitudinal motion control[2]. A combination of two ADAS is used to comply with this definition. Active cruise control regulates the vehicle to a predefined speed and slows down to maintain a preset distance with any slower moving vehicles ahead. Lane centering assistance (LCA) operates the steering system to track the trajectory computed by the automation (or AD trajectory), which typically is the center position of the lane in which the vehicle is traveling. Moreover, assistance for lane change is available in some vehicles. The automated lane change (ALC) function provides guidance to support the driver when the traffic condition is safe. Level-0 ADAS functions, such as automatic emergency braking for the longitudinal displacement or lane keeping assistance (LKA) for the lateral deviation complete the active safety envelope.

Providing an interactive environment with the steering system, where manual and automated inputs can coexist alleviates the risk of disengagement. Hence, lateral control of the vehicle is often shared so that manual steering over the guidance torque of the automation is possible without deactivation. Here, shared control is defined following[3]: 'human(s) and robot(s) are interacting congruently in a perception-action cycle to perform a dynamic task that either the human or the robot could execute individually under ideal circumstances. This definition excludes full automation (where there is no human) or manual control (where there is no automation)'. The concept of haptic shared control (HSC) has received significant attention due to the anticipated benefits to safety for partial and conditional automation levels[4–8]. Haptic communication through the steering interface is suggested to be the most practical channel to bond driver and vehicle because of its bilateral and dynamic characteristics[4,6,9].

Most partially automated vehicles use a blended control scheme for HSC (Fig. 1a). It finds its origin in robotic force control under the name of 'parallel force/position control'[10,11] and is based on the idea that the driver and the automation can apply a torque command independently to the same actuator. In terms of control, the automation is a feedback loop of the steering displacement, in which a manual torque input is seen as an external disturbance to be rejected. Conceptually, blended control consists in modulating the angle controller impedance to enable driver intervention. The ADAS functions are realized through conditional operation of the blended control scheme. Typically, the gain $G_t$ and eventually the angle controller gains are programmed to satisfy the operating condition of each ADAS function. For example, the assistance provided from the LCA function is obtained by operating the steering system in shared control mode with $0 < G_t < 1$. The reaction torque to the driver is proportional to the tracking error and its derivative. This angular error is caused by manual intervention or variation of the tracking reference. Therefore, the reaction torque of the LCA represents haptic guidance directed towards the AD trajectory, which is intended to reduce the driver workload.

If the surrounding traffic situation allows for a safe lane change, ALC is activated upon confirmation that the driver holds the steering wheel and activation of the turn indicator. The ALC consists in the application of a predefined trajectory change toward the adjacent lane center. When the lane change is completed, LCA is again activated to maintain the vehicle centered in the new lane.

LKA functions are often realized through brake activation to prevent lane departure. Torque vectoring is used to generate a vehicle yaw motion toward the lane center by applying asymmetrical commands to the individual brakes. This is a conservative approach that corrects the vehicle heading while reducing speed. In vehicles that employ the steering system for LKA, correction of the vehicle heading is achieved by adding a torque overlay[12].

As reported in the review of shared control for automated vehicles[8], there are more than 100 contributions focusing on shared steering control, thus revealing the wide range of applications and implementations of the HSC concept. Nevertheless,

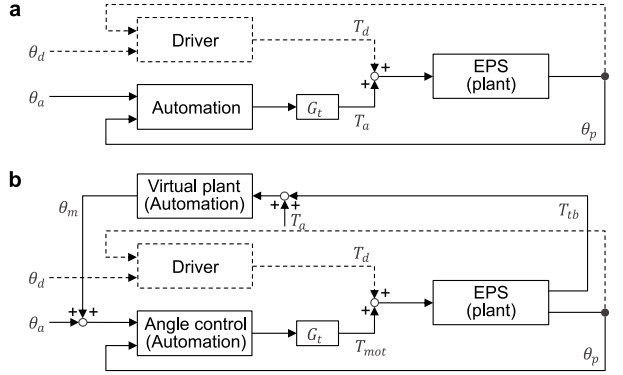

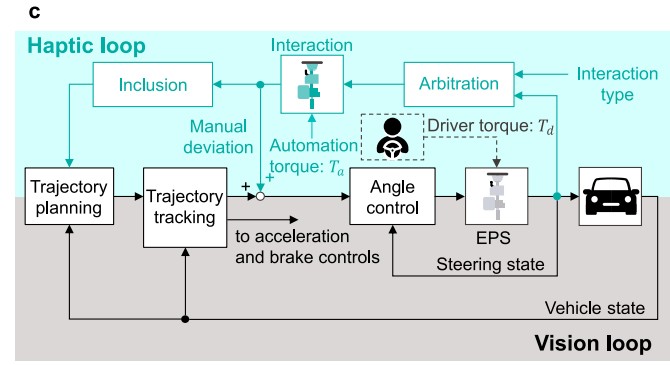

**Fig. 1 Two configurations of steering HSC between human driver and automation and overview of an automated driving control framework including the proposed collaborative steering control.** The dashed lines represent the human driver control. **a** Blended control: The driver torque $T_d$ and the automation torque $T_a$ track their target angles $\theta_d$ and $\theta_a$ with the feedback of the measured angle $\theta_p$. The respective tracking efforts are superposed to form an electric power steering (EPS) motor torque command. The gain $G_t$ is used for attenuating the automation effort and enabling driver intervention in shared mode. $G_t$ is set to zero when operating the EPS in manual mode in the event of an override. **b** Admittance control: A virtual plant is used to estimate the manual deviation $\theta_m$ from the measured driver torque input $T_{tb}$. The angle control attempts to enforce the superposition angle of $\theta_a$ and $\theta_m$ by applying the command torque $T_{mot}$ to the EPS. The reaction torque perceived by the driver is designed with the virtual plant and its load $T_a$. Similarly to blended control, the gain $G_t$ is set to zero for manual steering. **c** The black blocks with the vision loops illustrate the typical structure of an automated vehicle control system. The proposed control is represented with the turquoise blocks. Arbitration allocates the control authority of the automation based on the interaction type. The driver and the automation interact through the virtual EPS. The resulting manual deviation is input into the steering angle control so as to enforce the superposition of the driver intent to the AD trajectory. Additionally, this deviation is propagated to the inclusion block to assimilate the driver intent into the trajectory planning.

most contributions focus on particular issues, such as how to prioritize the driver versus the automation and how to manage conflict, therefore providing only limited answers toward a unified and holistic approach.

Further, state-of-the-art blended control has major disadvantages due to the dual role of tracking and regulation of the angle controller, which is typically a PID[13]. Ideally, perfect tracking is expected in the absence of driver input, while low rejection performance is required to enable manual intervention. Modulation of the control gain as a function of the driver activity is technically challenging because no sensor is available nor sufficiently reliable for this application (Supplementary Note 1).

Current practice is to consider each ADAS independently. This results in a discontinuous operation, which makes the driving experience uncomfortable. Consequently, drivers tend to display a low acceptance rate of ADAS technology[14]:

- LCA uses proportional and derivative gains independent of the driver input, while only the integrator is switched on to ensure zero steady-state error in the absence of driver intervention and switched off to avoid windup on manual input. Furthermore, the proportional and derivative gains are set to relatively low values to enable manual input, which lowers the tracking performance. Consequently, most partially automated vehicles have limited capability in tracking the lane in the case of road curvature. However, this centering torque is bounded by the driver input. Shared control is available under a preset driver torque threshold. Input above this threshold results in an override that deactivates the ADAS by returning the steering mode to manual ($G_t = 0$). When the driver torque decreases below the threshold, the ADAS is reactivated automatically by switching the steering mode back to HSC. Therefore, shared control for LCA is only available over a limited driver torque range resulting in discontinuous operation of the ADAS function[15].
- Although ALC provides comfort during regular operation, the steering operation is switched to manual in the case of driver intervention. The assistance interruption is uncomfortable and eventually requires driver reactivation[16,17].
- A torque overlay or offset is applied for LKA, which, in the worst case, results in the vehicle bouncing between the left and right lane markings. Shared control is not used because the low tracking performance of blended control does not guarantee lane tracking (as explained for LCA above) and therefore is not reliable for lane-keeping support in all road conditions (e.g., curves). While LCA and LKA share the control objective of centering the vehicle in the lane, they are not combined, increasing the risk of driver confusion.

Although technical limitations of mass produced cars justify some of these design choices, partially automated vehicles are characterized by limited functional integration of shared control and discontinuous operation of the ADAS[18]. Therefore, a generic control framework for collaborative steering, consistent across tactical and operational vehicle controls and across all levels of automation is required to address these issues.

In order to go beyond the classical form of driver-automation interaction this paper proposes a collaborative steering control framework within the limitation of mass produced steering hardware, that is based on the following functions:

- Interaction consists in providing the capability of haptic shared control to the steering system. Admittance control (Fig. 1b) is applied to enable the driver to deviate the vehicle from the AD trajectory without impairing the tracking performance of the angle control.

- Arbitration refers to the allocation of roles among the driver and the automation when attempting to share the lateral control of the vehicle. There are four types of interaction: cooperation, co-activity, collaboration and competition (see "Methods" section for their definitions). Based on a preselected type of interaction, an arbitration rule is used to set the reaction torque of the automation according to the motor control of the driver. The parameters of the driver motor control: goal (or target angle) and impedance, have to be estimated with the sensors available in mass produced vehicle.
- Inclusion consists in adapting the AD trajectory to the driver intervention. If the manual deviation is sufficiently large and persistent in time, the automation assimilates this correction in the trajectory planning.

Similarly to human-human collaboration[19,20], the above three functions are essential for the realization of collaborative steering. An interactive control environment is a prerequisite for haptic communication with the driver[21]. Arbitration provides the capability of interacting in different manners according to the road, traffic, and driver conditions. Then, inclusion assimilates the deviation resulting from the interaction into the AD trajectory. While the frequency bandwidth of the interaction has to be compatible with that of the driver torque, inclusion occurs at the lower bandwidth of the vehicle motion. There have been various attempts to provide human-robot collaboration, but none of them have combined the three functions of interaction, arbitration and inclusion. For example, the literature[22–26] addresses the interaction and arbitration problem following different approaches, but omit inclusion. Collaboration cannot be achieved because the robot does not assimilate the human intent in its trajectory. As soon as the human stops interacting, the robot returns to its predefined trajectory. Conversely, the literature[27–30] proposes adaption of the trajectory based on manual intervention (human force or torque) without arbitration. While driver triggered re-routing of the AD trajectory becomes available, it is performed at the relatively slow dynamics of the vehicle, which is inappropriate for haptic interaction. Indeed, fine-tuning of a vehicle steering behavior is a subjective process that defines the vehicle performance. Degraded steering feel caused by low frequency interaction control is unacceptable.

The main contribution of this work is the integration of the interaction, arbitration, and inclusion functions into a generic multi-level control framework that is applicable to mass-produced vehicles within the limitation of the available hardware. The control framework features the following advantages:

- Compatible with all levels of automation 0–4, where the human can still take part in the driving.
- Integration of the ADAS functions and continuous operation in shared control mode (override-free ADAS).
- ADAS functions that satisfy multi-objective requirements related to vehicle motion and driver intent to effectively contribute to better traffic safety.

Figure 1c gives an overview of the proposed control. The black blocks and the bottom half of the figure (gray background) represent the plant (detailed in "System dynamics" section) and the basic controls for automated driving. These controls are based on state feedback (positions and their derivatives) and rely on vision sensors (camera, radar, lidar, etc.). The turquoise blocks and the top half of the figure (light turquoise background) show the proposed control framework. The closed loop made with the arbitration and interaction blocks corresponds to the torque feedback of the admittance control illustrated in Fig. 1b and detailed in "Interactive steering control" section. The arbitration block allocates the control

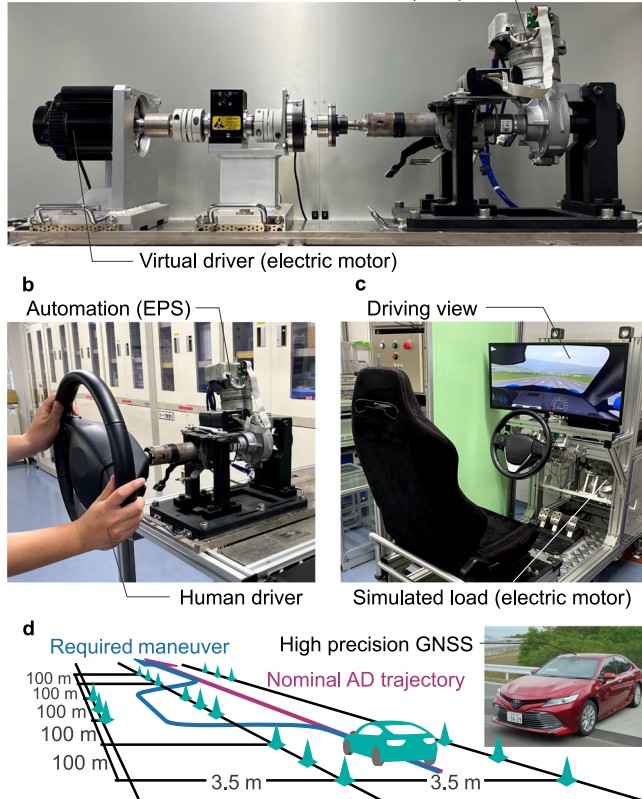

**Fig. 2 Test equipment. a** Virtual driver configuration. An impedance-controlled motor is used instead of the driver for the validation of the estimation of the driver motor control (driver goal and impedance). **b** Human driver configuration for the validation of the actual driver motor control and of the arbitration rules. **c** Driving simulator configuration for the validation of the trajectory adaptation. **d** Test vehicle configuration used for the proof of concept and the quantitative evaluation, and driving scenario for the quantitative evaluation. The following abbreviations are used: EPS for electric power steering, GNSS for global navigation satellite system, and AD for automated driving.

authority of the automation based on the estimation of the motor control of the driver ("Estimation of driver motor control" section) and on a preset type of interaction ("Arbitration" section). Propagation of the manual deviation resulting from the interaction to the trajectory planning is realized with the inclusion block ("Inclusion of driver intent into the trajectory adaptation" section), which closes the haptic loop.

Following the explanation of experimental configurations, the paper continues with the performance and experimental validation of the proposed control in the "Results" section. The "Discussion" section offers the contributions and limitations of the proposed control and the paper is concluded.

## Results

**Experimental configurations**. Four experiments have been conducted on different setups to test and validate the proposed control framework: virtual driver, human driver, driving simulator, and test vehicle (Fig. 2). These are summarized as follows and the symbols and parameters used for these experiments are listed in Supplementary Table 1 and Table 2.

- *Virtual driver configuration.* The first experimental configuration consists of a column-type EPS and an electric motor to replicate the driver input (Fig. 2a). Instead of the driver, an impedance-controlled motor is used for the validation of the estimation performance of the driver motor control ("Estimation of the driver motor control" section). The reference target angle and impedance of the virtual driver can be compared to their estimated values.

- *Human driver configuration.* The second setup uses the same equipment, but the impedance-controlled motor is replaced by a human driver (Fig. 2b). The human driver is required to execute a sine-shaped maneuver through the different preset types of interactions. The estimation of the driver impedance validated in the first test configuration is confirmed in the case of manual steering and used to verify the arbitration rules (Eq. (12)).

- *Driving simulator configuration.* The trajectory adaptation algorithm is validated on a static driving simulator (Fig. 2c). The control environment includes trajectory planning, tracking control, and the shared control framework. The vehicle motion is simulated and displayed on a screen for visual immersion. The driving scenario is a double lane change on a three-lane 1.5 km straight course. The nominal trajectory of the automation lies in the center of the middle lane and the vehicle is controlled to track this nominal trajectory at 60 km h$^{-1}$ using the Stanley trajectory tracking model[31]. The driver is required to operate the steering wheel only and is free to change lanes.

- *Test vehicle configuration.* This configuration concerns the implementation of the previously validated admittance control, arbitration rule, and trajectory adaptation in an actual test vehicle (Fig. 2d). The vehicle tracks a predefined trajectory (nominal AD trajectory) at 60 km h$^{-1}$ using cruise control on the same driving scenario and algorithm as that of the driving simulator configuration and position feedback from a high precision global navigation satellite system (GNSS). The driver is free to intervene and to deviate the vehicle away from the nominal AD trajectory. For the quantitative study ("Driver quantitative study" section), the driving scenario is the double lane change with 100 m intervals, as illustrated in Fig. 2d.

**Performance of the driver motor control estimation**. The performance of the estimation of the driver target angle (Eq. (15)) and impedance (Eq. (18) and Eq. (19)) were measured individually on the experimental configuration shown in Fig. 2a. For the impedance estimation, the goal of the virtual driver was set to a sine wave and that of the automation to zero. For the goal estimation, the impedance was set randomly, as shown in Fig. 3a. The estimation results are plotted in the same figure. The accuracy of the approximated driver goal varies as the driver impedance changes. When using the target angle of the virtual driver as input, the estimation of the driver stiffness and damping converge toward an oscillatory behavior about the set value. These oscillations stem from the driver impedance (Eq. (5)) that is undefined when either driver input or tracking error goes to zero[32].

Figure 3b shows the combined estimations under the same test conditions when the approximated driver goal is used as input for the estimation of the impedance. While the performance of the combined estimation is impaired, the impedance variations can still be extracted. Two major errors can be observed. First is the overestimation of the driver impedance in the steady-state conditions that is a consequence of the underestimation of the driver goal (Eq. (5)). In practice, the control is tuned for safe operation. Indeed, an overestimated impedance would amplify the role allocation from the arbitration rule (Eq. (12)). The second error is the amplification of the oscillatory behavior. This is caused because the model used in the extended Kalman filter

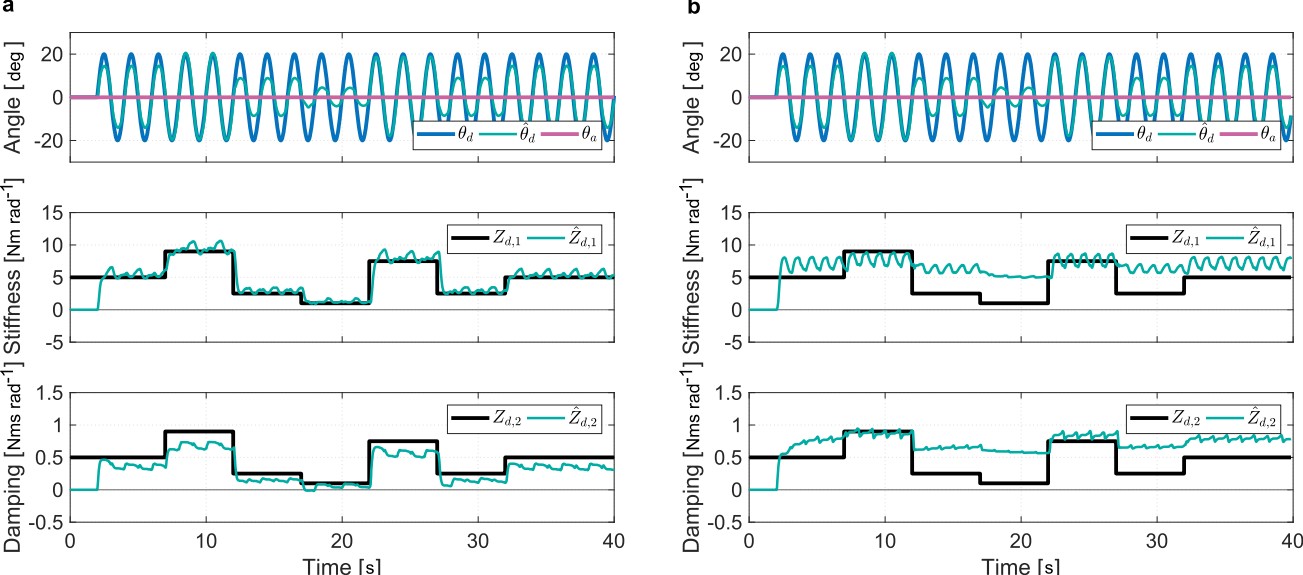

**Fig. 3 Independent and combined estimations of the driver motor control measured on the test bench (virtual driver configuration). a** The estimation of the driver target angle $\hat{\theta}_d$ is computed from the actual driver impedance $Z_{d,1}$ and $Z_{d,2}$, while the estimated driver impedance $\hat{Z}_{d,1}$ and $\hat{Z}_{d,2}$ are calculated with the actual driver target angle $\theta_d$. **b** The estimation of the driver target angle $\hat{\theta}_d$ is computed from Eq. (13) and the estimated driver impedance $\hat{Z}_{d,1}$ and $\hat{Z}_{d,2}$ are calculated with the estimated driver target angle.

(EKF) is different from that used for the approximation of the driver goal. This class of oscillatory problems caused by modeling error is well-known[33].

**Verification of the arbitration rule**. Setting the automation effort with the arbitration rule according to a type of interaction (Eq. (12)) is verified in this experiment. The human driver configuration shown in Fig. 2b is used so that the human can take part in the experiment. The driver was asked to perform a steady slalom maneuver while the automation had the objective of driving in a straight line (Fig. 4). The type of interaction was changed every fifteen seconds in the following order: (i) co-activity, (ii) collaboration, and (iii) competition. Furthermore, the driver was asked to take his hands off the steering wheel during the last five seconds of each interaction type. The automation impedance is set to be constant ($\kappa = 0$) for co-activity during the first 15 s. The measurements show that the pinion angle tracks the average angle between those of the driver and the automation in this particular case, where the driver accommodates the automation. Since the automation impedance is constant, the driver torque is simply proportional to the angular deviation from the automation target. During the next fifteen seconds, $\kappa = 1$, which corresponds to collaboration. The automation impedance is adapted based on the estimated driver impedance: the larger the driver impedance, the smaller the automation impedance. During the manual intervention, the driver perceives resistance from the higher authority of the automation at first. Then, as the automation detects driver engagement, the control authority is gradually transferred to the driver. Conversely, when the automation detects that the manual intervention fades, the automation impedance is recovered and the automation target angle is tracked. This demonstrates how automation backs up the human in the driving task with a continuous estimation of the driver motor control. From 30 s onward, $\kappa = -1$ which sets the competition type of interaction. The automation impedance increases according to that of the driver in order to oppose manual intervention. The driver has to apply higher torque to accomplish the same maneuver. Smaller values of $\kappa$ enable stronger resistance and virtually full rejection of the driver intervention.

During the three time periods in which the driver is not holding the steering wheel (hands-off), the control authority is naturally returned to the automation (nominal impedance). This highlights that the automation works as a backup to the driver but also that sustained effort is required for any deviation away from the AD trajectory. Automation backup is suitable for automated driving level 3 or more but not at level 2, where the driver is required to be engaged in the driving task, as it may increase the risk of misuse.

**Performance of the trajectory adaptation**. This section summarizes the results obtained for the trajectory adaptation on the static driving simulator, shown in Fig. 2c. Co-activity role allocation ($\kappa = 0$) is chosen to focus on the trajectory adaptation without the estimation of the driver motor control. The measured torque, the AD trajectory, and the actual vehicle trajectory are compared when the adaptation is deactivated and activated for a double lane change maneuver (Fig. 5). When deactivated, the AD trajectory is fixed on the initial lane and the driver has to continuously apply torque to deviate the vehicle away from the AD trajectory (Fig. 5a). The double lane change maneuver is performed at the expense of sustained effort as the automation continuously pulls the driver back to the AD trajectory. This interaction is of interest because it provides guidance to the driver. While large deviation may result in high interaction torque, it is fundamental as a haptic cue during local deviation.

As the trajectory shifts towards the next lane with the adaptation algorithm activated, the interaction torque relaxes and the vehicle is centered on that new lane (Fig. 5b). The driver applies torque to initiate a local deviation, which triggers an adaptation of the trajectory if sufficiently large. The bounded interaction torque and the adaptive guidance constitute the relevant haptic cues for collaborative steering.

**Proof of concept on the test vehicle**. The previously validated arbitration and inclusion control algorithms were implemented on the test vehicle shown in Fig. 2d. The driving scenario is the same double-lane change as that in the previous section and both

arbitration and inclusion controls were active. Two responses are provided for the arbitration rule set to collaboration (Fig. 6a) and to competition (Fig. 6b). The practical verification of the

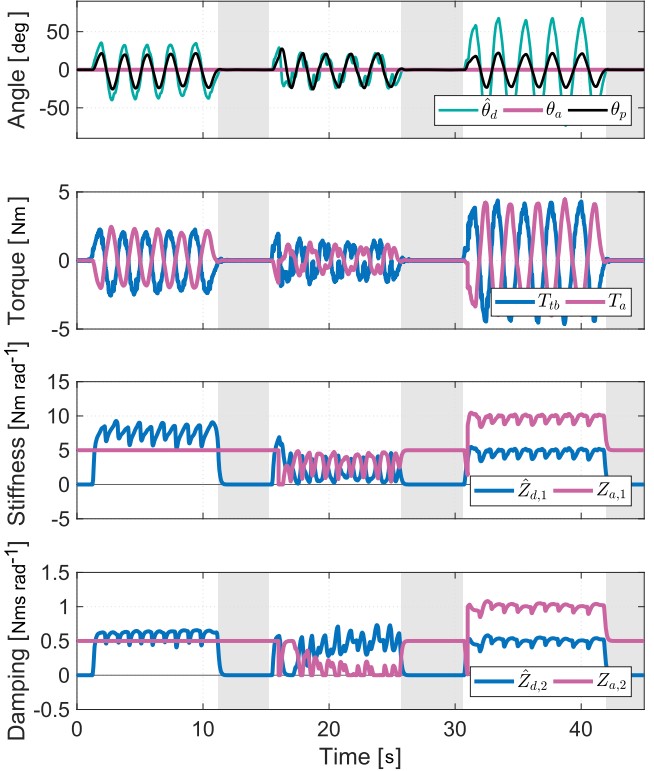

**Fig. 4 Interaction performance for different types of interaction measured on the human driver configuration.** The estimation of the target angle $\hat{\theta}_d$ is computed from the road information and the measured driver torque $T_{tb}$ with Eq. (13), and the estimated driver impedance $\hat{Z}_{d,1}$ and $\hat{Z}_{d,2}$ is calculated with the estimated driver target angle $\hat{\theta}_d$. Based on the estimated driver impedance $\hat{Z}_{d,1}$ and $\hat{Z}_{d,2}$ and a preselected type of interaction, an arbitration rule (Eq. (12)) is used to modulate the automation impedance $Z_{a,1}$ and $Z_{a,2}$, which finally generates the automation torque $T_a$ to track its target angle $\theta_a$ with the feedback of the measured angle $\theta_p$. The type of interaction is set to co-activity ($\kappa = 0$) from 0 to 15 s, to collaboration ($\kappa = 1$) from 15 to 30 s, and to competition ($\kappa = -1$) from 30 s onward. The sections with the gray background indicate time periods where the driver has his/her hands off the steering wheel.

proposed multi-level haptic control demonstrates a consistent response of the test vehicle. First, the driver interacts with the automation under the control allocation set with the arbitration rule. Because the driver and the automation impedances are complementary in collaboration mode, the torque peak that is observed is lower than that during co-activity (Fig. 5b). Second, the lateral deviation induced from the interaction is propagated to the trajectory planning via the inclusion control. Consequently, the vehicle tracks the left and right lanes without sustained manual torque.

In competition mode, the automation impedance varies together with that of the driver. Because of the relatively low value of $\kappa$, the driver cannot apply a torque high enough to deviate the vehicle from the AD trajectory without reaching the maximum capacity of the steering system. No assimilation of the manual intervention is verified, and the vehicle tracks the trajectory developed without haptic contribution.

In summary, the haptic cue communicated to the driver is twofold. First, the driver feels the set role allocation and may be able to generate a lateral deviation of the vehicle. Second, if this deviation is large enough, inclusion occurs by means of haptic adaptation of the trajectory. As a consequence, the interaction torque remains bounded, indicating to the driver that his intervention has been assimilated. These haptic cues contribute to the realization of intuitive collaborative steering.

**Driver quantitative study.** This section presents an evaluation of several individuals to verify the usefulness of the proposed multi-level haptic control framework. Five participants with an average age of 35 (from 29 to 44) years old took part in the driving assessment on the test vehicle shown in Fig. 2d. All participants were experienced drivers and reported an average annual travel distance of 5800 km. The participants were required to execute double lane change maneuvers at 60 km h$^{-1}$, as illustrated in Fig. 2d, with the four different control modes (Table 1) set in random order. All participants drove twice under each control mode and the averages of these trials are used for this study.

Two criteria are proposed for the assessment of collaborative steering: driver effort (DrE) and steering entropy (StE). DrE corresponds to the driver torque steering effort throughout the test duration $t_{sc}$[28,34], while StE is a criterion representing the smoothness of the evolution of the steering angle that is commonly used to quantify maneuverability[35,36]. Their respective formulations are given in the "List of KPIs" section. Control modes with lower DrE and StE allow for a smoother operation

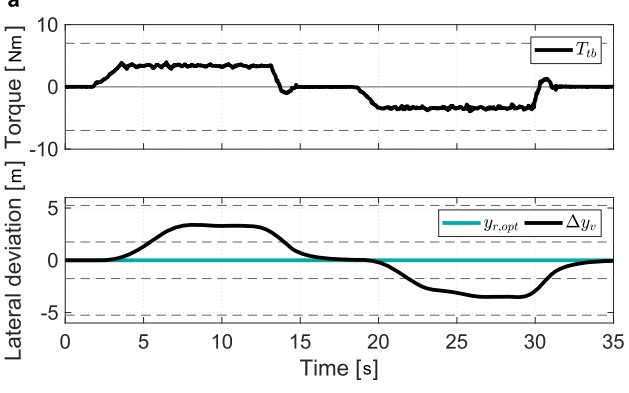

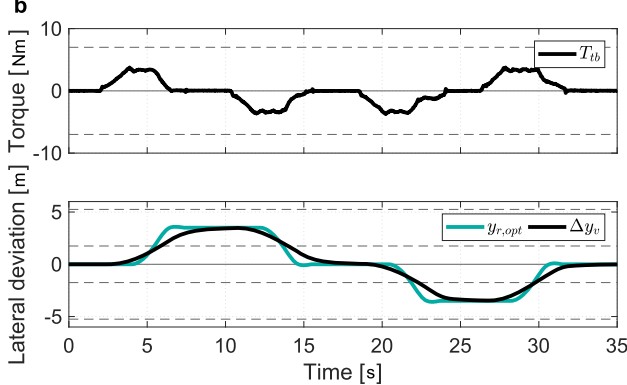

**Fig. 5 Comparison of driver inputs over a double lane change maneuver with the trajectory adaptation inactive and active measured on the driving simulator (driving simulator configuration).** The starting of the lane changes correspond to the points where the measured driver torque $T_{tb}$ arises. **a** The automation (AD) trajectory $y_{r,opt}$ is not adapted, and the driver and the automation track the actual vehicle trajectory $\Delta y_v$. **b** The AD trajectory $y_{r,opt}$ is adapted according to the driver intervention, and the driver and the automation track the actual vehicle trajectory $\Delta y_v$.

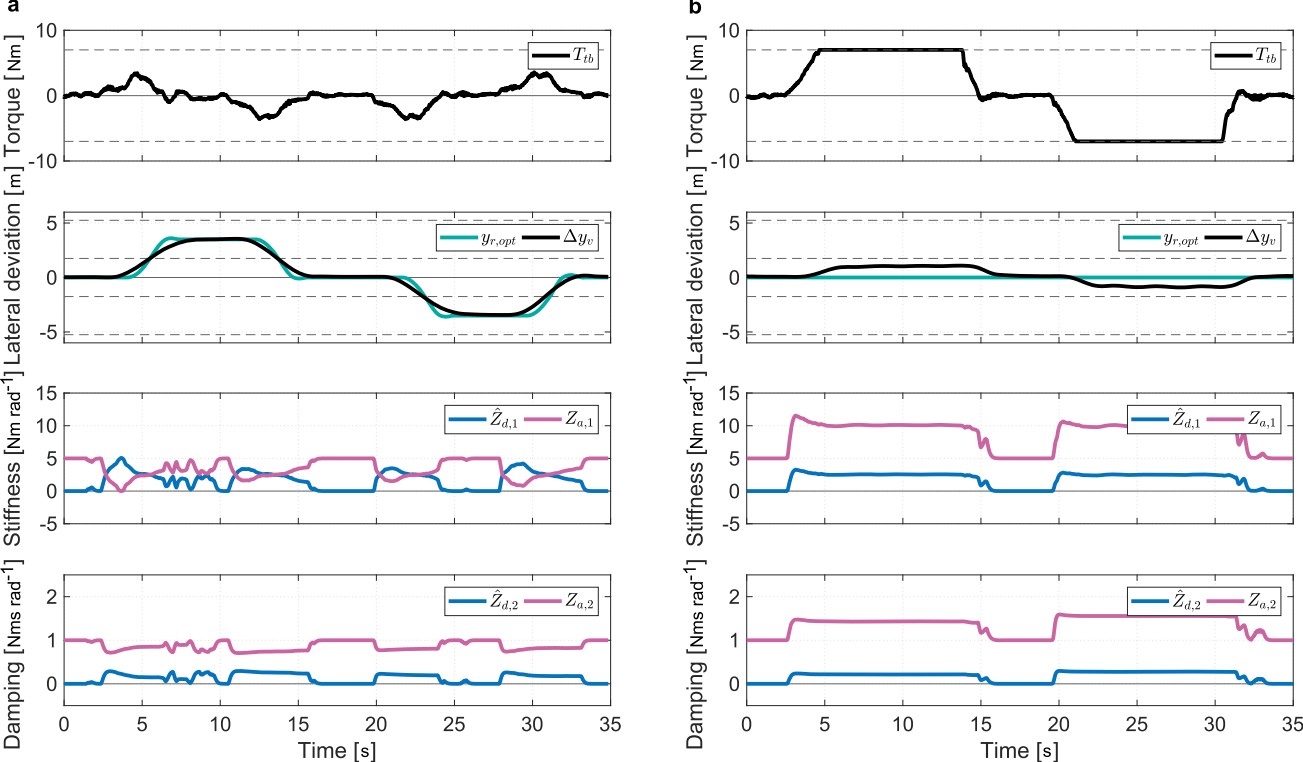

**Fig. 6 Performance of the complete multi-level haptic control measured on the test vehicle (test vehicle configuration).** All symbols are compatible in Fig. 3–Fig. 5. **a** The type of interaction is set to collaboration ($\kappa = 1$). **b** The type of interaction is set to competition ($\kappa = -2$).

| Table 1 Four control modes for the quantitative study | | | |
| --- | --- | --- | --- |
| Mode no. | Interaction type | $\kappa$ | Inclusion |
| 1 | Co-activity | 0 | Inactive |
| 2 | Collaboration | 1 | Inactive |
| 3 | Co-activity | 0 | Active |
| 4 | Collaboration | 1 | Active |
| $\kappa$ is the parameter used to set the type of interaction. | | | |

with less effort for the driver. Statistical differences between the control modes were analyzed using one-way analysis of variance (ANOVA), and multiple comparisons between specific control modes were executed via paired samples t-tests.

The results obtained with each control mode are summarized in Fig. 7. Figure 7a shows that the DrE significantly decreases in the order of modes 1–4 ($p < 0.001$) according to the ANOVA results. In particular, there is a large gap in DrE under modes 3 and 4 compared to modes 1 and 2. This is interpreted as the result of the implementation of inclusion, where the AD trajectory was adapted to match the driver intention so that sustained manual torque was no longer required during the lane change maneuver.

Further, Fig. 7b suggests that the application of arbitration reduces the average of StE both with and without inclusion, i.e. the maneuver was executed more smoothly. In particular, the lowest StE is obtained under mode 4, compared to mode 1 ($p < 0.04$) and mode 3 ($p < 0.02$), according to the t-test results. StE shown in Fig. 7b suggests that the variability between participants was higher than that of DrE (Fig. 7a), especially under modes 1 and 2. In addition, the participants can be classified into two groups: group 1 includes participants 1 and 2, and group 2 includes the others. The StE is smaller and the DrE larger in group 1 compared to group 2 (Fig. 7c, d). From these

trends, it can be inferred that participants of group 1 applied more effort to achieve a smooth maneuver, while participants of group 2 operated with a smaller torque input at the expense of smoothness. These variations in driver behavior suggest that control modes 1 and 2 may lead to a low rate of acceptance, whereas control modes 3 and 4, which yield a smaller variability, are likely to be accepted by a wide range of drivers.

The comprehensive analysis of these two criteria (DrE and StE) suggests that the proposed control framework based on arbitration and inclusion has the potential to achieve smooth maneuvering with less effort for a wide variety of drivers.

## Discussion

The proposed control framework enables collaborative steering through haptic control integration at the operational and tactical levels of automated driving vehicle. A broad spectrum of interactions between the driver and the automation is made available through the arbitration rules, and manually induced deviations are consistently assimilated and updated via trajectory planning.

Compared to the literature[22–26], which consider interaction and arbitration only, the proposed control framework prevents the vehicle from returning to the nominal AD trajectory after a driver intervention. Inclusion assimilates intervention as an additional factor alongside vision information for the rerouting of the AD trajectory (Fig. 5b and Fig. 6a). This enables maintaining continuous shared steering operation in the event of a manually induced lane or route change. For example, by tuning the inclusion parameters and selecting the appropriate interaction type, LCA, ALC, and LKA can be integrated consistently in partially automated vehicles.

Compared to control algorithms that consider solely interaction and inclusion[29,30], the proposed framework permits full rejection of the driver input by setting the interaction type to competition (Fig. 6b). This means that it can accommodate any

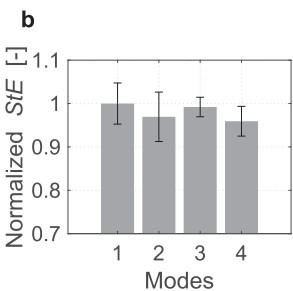
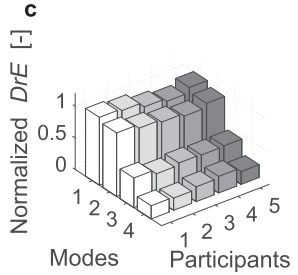
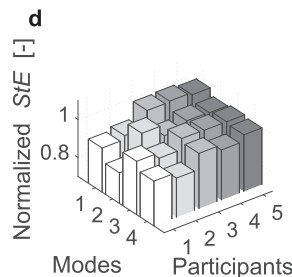

**Fig. 7 Quantitative evaluation by driver effort (DrE) and steering entropy (StE) measured on the test vehicle (test vehicle configuration).** DrE and StE are normalized with the min–max normalization method where the maximum value is the mean of mode 1 and the minimum value is zero, i.e. the lower bound of the physical range. The error bars represent the standard deviations between participants. **a** DrE evaluation between control modes. **b** StE evaluation between control modes. **c** DrE evaluation between control modes and participants. **d** StE evaluation between control modes and participants.

level of automation and the development of multi-objective ADAS functions. It is the arbitration that provides this capability (Fig. 7). For partially automated vehicles, the LKA function can be enhanced with a high temporary reaction torque to prevent a collision. Also, this applies to highly automated vehicles where the automation can take the responsibility of the OEDR. The automation will have the authority to collaborate or compete with the driver depending on the road and traffic situation.

Compared to control schemes that merely rely on inclusion[27,28], the proposed control framework enables independent optimizations of the driver and vehicle responses. Driver reaction torque is tuned with the interaction and arbitration functions, while the vehicle motion is adapted with the inclusion function. This alleviates the tuning trade-off and results in higher overall performance.

With the admittance control, the trade-off between the acceptance of the driver input and the tracking accuracy, which is found in blended control, is solved, and the tuning range is widened significantly.

The implemented framework requires the interaction type between the driver and the automation to be set by a higher-level controller based on endogenous (driver state) and exogenous (road and traffic conditions) information. Since the selection of the interaction type is out of scope for this work, the appropriate interaction type setting according to the driving situation remains to be addressed in a future task.

The adaptation of the automation impedance based on the preset type of interaction is simplified with Eq. (12) in comparison to the optimization method[37] (Fig. 4). This approach allows the validation of the comprehensive concept of arbitration while satisfying the implementation requirement on mass production hardware. However, to faithfully realize the interaction types originally defined in the literature[37], a control theory to minimize the cost function consisting of the angle tracking error and effort of the driver and the automation defined for each type of interaction is required. In a future task, this could be achieved by using non-linear model predictive control (MPC)[25,26].

Although the accuracy of the driver goal approximation used for arbitration is limited, it proves to be sufficiently rich to extract the dynamics of the driver impedance (Fig. 4). Moreover, the implemented approximation method, which merely relies on an admittance model, requires relatively low computational power. Nevertheless, the driver goal and impedance are abstract concepts, and it has not been verified whether the values estimated by the current algorithm match the actual driver motor control. However, this is a conceptual proposal to roughly approximate how strongly the driver is steering the vehicle, in order to implement the arbitration strategy. A further limitation is that the EKF tuning for the driver impedance estimation is based on the

assumption that the stiffness and damping of driver change simultaneously. This means that the EKF cannot capture situations where the driver operation causes an extreme change only in his stiffness or damping. This limitation could be improved by the comparison with measured driver operation information captured via electroencephalography (EEG) or electromyography (EMG) sensors.

Using the manual angle as input to the vehicle model to estimate the yaw rate is robust to modeling errors compared to using the driver torque as suggested in the literature[27,28]. Furthermore, as the manual deviation is related to the type of interaction, propagation of this deviation to the trajectory planning enhances haptic consistency. Vehicle tests demonstrated the capability of interacting under the role allocated by the arbitration and the inclusion to manual intervention. However, the timing for that propagation from the initial manual intervention to the trajectory adaptation should be carefully adjusted to guarantee an acceptable steering feel. Hence, further fine-tuning and customization of the proposed control strategy is essential for intuitive haptic communication and driver acceptance.

The analysis of DrE and StE suggests that the proposed control framework (Mode 4 in Table 1) can achieve smoother maneuvers with less effort for a wide variety of drivers compared to controls that use arbitration only (Mode 2 in Table 1)[22–26] or which consider solely inclusion (Mode 3 in Table 1)[27–30]. However, since the test samples are relatively small, it would be worthwhile to validate the proposed control with a larger number of participants to obtain a quantitative evaluation of greater statistical relevance.

## Conclusion

A driver-centered automation control has been proposed to address the concept of collaborative steering in automated driving without alteration of the hardware available in mass-produced vehicles. According to a preset type of interaction, the driver steering intention is reflected in the automation impedance and trajectory planning. Because the implication of manual intervention affects both operational and tactical levels of automated driving control, intuitive haptic communication is made available to the driver and consistent integration across all vehicle actuators is supported.

The originality of the proposed implementation is summarized as follows:

- The proposed multi-level control framework enables consistent integration of the ADAS functions while continuously operating in shared control mode. Furthermore, the high-performance angle control, combined with the large spectrum of interaction, makes this framework

compatible with all automation levels where the driver can still be part of the driving.

- The admittance control has been applied to a steering system to enable interaction between driver and automation. The interactive nature of admittance control alleviates the trade-off found in blended control while ensuring superior tracking performance of both AD trajectory and driver intervention. Furthermore, the interactions taking place in the virtual plant are isolated from hardware limitations resulting in robust performance.
- A large spectrum of interaction between independent agents has been made available with the proposed rules of arbitration.
- Consideration of the context of collaborative steering enables the assumptions of independent interacting agents. The observability issue of the combined estimation of driver goal and impedance is avoided by considering the agent goals as boundary conditions and consequently, impedance modulation can be achieved.
- Practical reconsideration of classical two-level steering control within the context of collaborative steering resulted in the development of a simple approximation of the driver goal.
- The manual deviation from the AD trajectory resulting from the interaction is consistently propagated to the trajectory planning by using the manual angle as input.
- Through quantitative evaluation with five participants, the proposed multi-level haptic control has been validated in a vehicle. The assessment suggests a significant potential to provide smooth collaborative steering with less effort for a wide range of drivers.

While the proof of concept on the test vehicle demonstrates the capability of this multi-level collaborative steering control, fine-tuning and customization is required to render the steering feel comfortable and consistent for a safer and more reliable shared driving experience.

Finally, the application of the proposed control framework for the development of ADAS functions can be considered with the objective of encouraging driver engagement at partial automation or providing continuous automation back up to the driver at higher automation levels.

## Methods

**System dynamics**. The system enabling collaborative steering is composed of the driver, the automation, and an electric power steering (EPS) system, which represents the mechatronic interface. The EPS is composed of a steering wheel, a motor, gears, and angle and torque sensors, as shown in Fig. 8. The dynamics of the

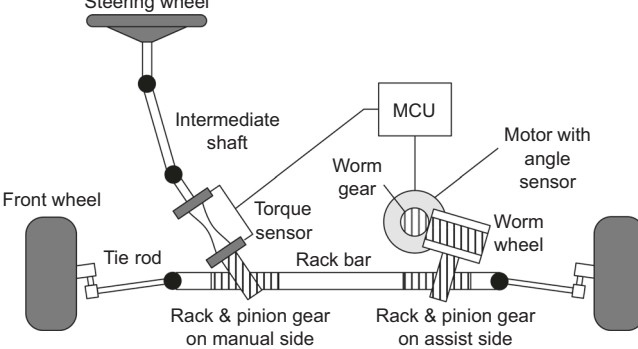

**Fig. 8 Structure of a dual pinion type electric power steering.** In manual operation, the motor is controlled so that less effort is required for the driver when turning the wheels. For collaborative steering, the automation, which input is computed in the motor control unit (MCU), is controlled so as to support appropriately the driver.

EPS system can be described as:

$$T_d + i_s T_{mot} + \epsilon = \Psi \tag{1}$$

where $T_{mot}$ is the motor torque command, $T_d$ is the driver input torque, $i_s$ is the ratio of the reduction gear, $\epsilon$ is white noise in the driver and the automation torque, and $\Psi$ is the dynamics of the EPS. Assuming that the components from the lower side of the torque sensor to the front wheel are stiff, $\Psi$ can be simplified to a two-inertia system[38]:

$$J_{sw}\ddot{\theta}_{sw} = T_d - T_{tb} \tag{2}$$

$$T_{tb} = K_{tb}(\theta_{sw} - \theta_p) \tag{3}$$

$$J_p\ddot{\theta}_p = T_{tb} + i_s T_{mot} + T_{ld} \tag{4}$$

where $J_{sw}$ and $J_p$ are the steering wheel and the lower part of the torque sensor inertia, respectively, $\theta_{sw}$ and $\theta_p$ are the steering wheel and the measured pinion shaft angles, $T_{tb}$ and $K_{tb}$ are the torque sensor output and its stiffness, and $T_{ld}$ is a disturbance consisting of internal nonlinearities (friction, backlash, etc.) and the road load.

Both agents, the driver, and the automation are assumed to track their own trajectory based on individual impedance control loops. The motor control of the driver holding the steering wheel is formulated as follows:

$$T_d = -Z_d'\xi_d, \; Z_d = \begin{bmatrix} Z_{d,1} \\ Z_{d,2} \end{bmatrix}, \; \xi_d = \begin{bmatrix} \theta_{sw} - \theta_d \\ \dot{\theta}_{sw} - \dot{\theta}_d \end{bmatrix} \tag{5}$$

where $Z_d$ is the driver impedance ($Z_d \in \mathbb{R}^2_{\geq 0}$), $\xi_d$ is the tracking error of the driver, and $\theta_d$ is the target angle or goal of the driver. $'$ represents a transpose matrix.

The effort $T_a$ is the torque input of the automation in Fig. 1b. It constitutes one of the components of the EPS motor torque $T_{mot}$ (see the next section):

$$T_a = -Z_a'\xi_a, \; Z_a = \begin{bmatrix} Z_{a,1} \\ Z_{a,2} \end{bmatrix}, \; \xi_a = \begin{bmatrix} \theta_p - \theta_a \\ \dot{\theta}_p - \dot{\theta}_a \end{bmatrix} \tag{6}$$

where $Z_a$ is the automation impedance ($Z_a \in \mathbb{R}^2_{\geq 0}$), $\xi_a$ is the tracking error of the automation, and $\theta_a$ is the target angle or goal of the automation.

To represent how the driver and the automation impedances may evolve over time, the following dynamic models are introduced[39].

$$\dot{Z}_d(t) = -T_{z,d}^{-1}Z_d(t) + T_{z,d}^{-1}Z_d(t-1) \tag{7}$$

$$\dot{Z}_a(t) = -T_{z,a}^{-1}Z_a(t) + T_{z,a}^{-1}Z_a(t-1) \tag{8}$$

where $T_{z,d}$ and $T_{z,a}$ are time-constant parameters for modulating the driver and the automation impedances.

**Interactive steering control**. The interpretation of "physical human-robot interaction" (pHRI) has significantly evolved over the past decades. While safety was originally the main concern in the case of physical contact with a robot, pHRI has been considered as an implicit means to communicate the human intention to a robot with the objective of jointly completing a task[40]. The literature[41] groups control strategies for pHRI into two categories: "indirect force control" and "direct force control". The former controls the force through motion feedback, with typical applications of impedance and admittance controls. The latter has the objective of controlling the interaction force to the desired value based on the feedback from the actual force measurement. The objectives of the interactive control of the steering actuator are twofold:

- High-angle tracking performance
- Enabling manual deviation from the AD trajectory without impairing the angle-tracking performance

An admittance control framework (Fig. 9a) is proposed for the interactive steering control to overcome the limitations of blended control. Although admittance control is not commonly used for haptic interaction[42], it is appropriate for the application of automated steering because of the high performance of position tracking and the availability of the measurement of the driver torque. Assuming that a lower-level controller linearizes and decouples the plant dynamics, a linear two-degree of freedom controller (feedback and feedforward) with a single set of gains is sufficient to guarantee constant position tracking performance under any operating condition. One of the advantages of admittance control is that the inner angle control loop is purposefully made stiff so as to ensure high tracking performance. Consequently, the AD trajectory is tracked accurately in the absence of interaction. Conversely, the outer torque loop is naturally closed in the presence of interaction[43]. The reference position of the automation $\theta_a$ is corrected with an estimated manual deviation $\theta_m$ computed from the dynamics of the virtual plant.

$$J_{vp}\ddot{\theta}_m = T_{tb} + T_a \tag{9}$$

The angle reference of the inner loop $\theta_{cmd}$ is defined as the superposition of

**Fig. 9 Detailed representation of the admittance control structure for haptic shared control and equivalent interaction dynamics of an admittance-controlled electric power steering (EPS). a** The dashed lines represent the driver control. The inner loop is an angular position control, which is purposefully made stiff. The outer loop is activated only when the driver inputs torque. The virtual EPS computes an estimation of the manual deviation, which reflects the driver intent under the preset type of interaction. The manual deviation is superposed to the angular command of the automation (AD trajectory) to form the command of the inner loop. **b** The equivalent interaction dynamics is a two-inertia system coupled with the torque sensor, which stiffness is $K_{tb}$. The steering wheel (inertia $J_{sw}$) represents the interface with the driver motor control (goal $\theta_d$ and impedance $Z_d$), while the reaction from the automation (goal $\theta_a$ and impedance $Z_a$) is applied to the virtual inertia $J_{vp}$.

commands from the automation and the driver:

$$\theta_{cmd} = \theta_a + \theta_m \tag{10}$$

Hence, the automation angle control attempts to enforce the angle superposition of $\theta_a$ and $\theta_m$ by applying the motor torque command $T_{mot}$.

The closed-loop system dynamics are obtained by substituting Eq. (2), Eq. (5), and Eq. (6) into Eq. (9) and assuming perfect tracking ($\theta_{cmd} \approx \theta_p$).

$$-J_{sw}\ddot{\theta}_{sw} - Z'_d \xi_d = J_{vp}\ddot{\theta}_m + Z'_a \xi_a \tag{11}$$

This equivalent two-inertia system is illustrated in Fig. 9b. It shows that the torque felt by the driver ($T_d = Z'_d \xi_d$) when interacting with the automation can be controlled by tuning the virtual plant and the automation effort ($T_a = Z'_a \xi_a$). For stability reasons, the bandwidth of the outer torque control loop should be set lower than that of the inner angle control loop[42–44]. In practice, the inertia of the virtual plant is set to a value higher than that of the actual plant ($J_{vp} > J_p$). Hence, it is the automation effort that is modulated to render the interaction. As shown in the next section, an arbitration rule is used to allocate the automation control authority according to a preset type of interaction.

In consequence, the admittance control framework enables manual deviation $\theta_m$ of the vehicle from the AD trajectory $\theta_a$. When the manual intervention ends ($\theta_m = 0$), the steering returns back to the AD trajectory ($\theta_{cmd} = \theta_a$).

**Arbitration**. Arbitration in pHRI is required to regulate the control authority of the robot when attempting to accomplish a common task according to a preset type of interaction. The literature[37] proposes a taxonomy of the types of interactions based on neuroscience and game theory:

- Assistance is an extreme case of cooperation where, typically, the robot (slave) is used to amplify the physical capability of the human (master).
- Cooperation takes place when the two agents work towards a common end and need each other to reach the goal. Part of cooperation is the education role arbitration, which is critical for gaining new capability by ensuring a certain degree of engagement.
- Co-Activity occurs when the two interacting agents, without knowledge of each other's actions, incidentally succeed in a common task.
- Collaboration features no fixed roles distribution but rather adapts the distribution to accommodate the other while still considering its own perspective.
- Competition, similarly to collaboration, is a symmetric arbitration where the role distribution opposes the other while considering its own perspective.

Review[40] cites numerous contributions for each interaction type. The type of interaction is likely to vary dynamically over the completion of a joint task. Endogenous (driver state) and exogenous (road and traffic conditions) information is used in a higher-level controller to set the interaction type, which is, however, outside the scope of this work.

The objective of the arbitration is to define how the automation has to react to the driver intent based on the preset type of interaction. From Eq. (11) and assuming constant virtual plant inertia, two variables, the automation angle $\theta_a$, and its impedance $Z_a$, are available for adjusting the reaction torque as a function of the driver goal $\theta_d$ and impedance $Z_d$. Two approaches have been proposed in the literature[24] focuses solely on the interaction and avoids consideration of the boundary conditions by opting for constant human and automation impedances. In this way, an arbitration rule was established based on the human goal and resulted in a large spectrum of interaction but with limited dynamic performance. Conversely, the literature[23] addresses the application of robotic rehabilitation, which relies on cooperation, with the human goal assumed to be equivalent to that of the robot. Under this assumption, impedance

modulation was developed for this specific type of interaction. Here, the proposed arbitration considers that driver and automation are two independent agents. Therefore, their respective goals are boundary conditions that need to be identified separately. Then, the driver impedance can be estimated with the EKF when knowing the driver goal (detailed in the next section).

Here, the following arbitration rule is proposed for the adaptation of the automation impedance:

$$Z_a = Z_{a,0} - \kappa \hat{Z}_d \tag{12}$$

where $Z_{a,0}$ is the nominal automation impedance, $\hat{Z}_d$ is the estimated driver impedance, and $\kappa \in \mathbb{R}$ is a parameter used to set the type of interaction. For $\kappa = 0$, the automation impedance is constant, which corresponds to co-activity. This is the natural type of interaction obtained from the admittance control. For $\kappa > 0$, the automation adapts and supports the driver. This is the collaboration type of interaction. The opposite behavior or competition is obtained for $\kappa < 0$. Here the automation impedance increases with that of the driver, resulting in a rejection of the manual intervention. With this approach, the range of interactions from competition to co-activity and collaboration is made available. However, cooperation (including assistance) is not applicable because of the assumption made regarding the independent goals of driver and automation. However, note that cooperation-type interaction is already being used in the EPS control for manual operation: the EPS (automation) amplifies the manual torque so as to assist the driver in reaching their goal.

**Estimation of the driver motor control**. Realization of the arbitration relies on the availability of the driver goal and impedance. The observability issue of a combined estimation of these two variables with the interaction dynamics[32] (Eq. (11)) is avoided with independent estimations. Indeed, it is assumed that the contextual nature of the joint task of driving defines the boundary conditions (driver and automation goals) of the interaction dynamics. Then, the driver and the automation vary their impedances as they interact under the constraints of their respective goals.

The driver goal and impedance are abstract representations of how the driver interacts with the automation. Although numerous driver models have been proposed, their objectives are to represent the driver under particular conditions. These objectives range from vehicle tracking of a given trajectory with a virtual driver model to more elaborated driver models, which include trajectory planning with optimization preference (time, acceleration, braking, rpm, etc.)[45]. However, there is no practical and generic approach available that could predict where a driver intends to go in any situation. Similarly, various attempts to describe and identify the driver impedance have been proposed but they rely either on additional sensors (e.g. EMG, grip force, driver torque) or are laboratory-based setups with limited practical relevance[46,47]. The literature[48] proposes the identification of the driver impedance while driving under the assumption of a constant driver goal. Unfortunately, these approaches are not suitable for the estimation of the driver impedance while interacting with the automation.

Considering the economic constraints of mass-produced vehicles with a limited number of sensors available, the driver goal and impedance can, at best, only get approximated roughly. Here, an approximation of the driver goal is computed at first. The sensors available in mass-produced vehicles are limited, so the two-level model of steering[49] is applied in the context of collaborative steering[7]. The driver anticipatory visual open-loop control is assumed to track the center of the lane as an inherent environmental constraint. Any deviation from it is considered as originating from a driver intent in the compensatory closed-loop control. Hence, the estimate of the driver goal $\hat{\theta}_d$ is composed of an environmental constraint $\theta_{env}$ and of the driver intent $\theta_{int}$.

$$\hat{\theta}_d = \theta_{env} + \theta_{int} \tag{13}$$

A steady-state model of the vehicle motion with longitudinal speed $v_x$ and road

**Fig. 10 Vehicle and constant turn ratio and velocity (CTRV) model. a** Representation of the single track vehicle model used for the calculation of the yaw rate $\gamma_m$ from the manual deviation $\theta_m$. $\beta$ is the side slip angle, $v_x$ is the longitudinal velocity, and $i_o$ is the overall gear ratio from the steering angle to the tire angle $\delta_m$. $l_f$ and $l_r$ are the distance from the gravity center to the front and rear axles, respectively. **b** CTRV model for the calculation of the driver desired lateral deviation $\Delta y_d$ when the vehicle moves with the constant yaw rate $\gamma_m$ and longitudinal velocity $v_x$ during a time horizon $t_s$. Representation of the lateral deviation caused by the manual intervention is made in the Frenet coordinate. The AD trajectory is represented on the s-axis and any deviation from it corresponds to a relative displacement along the d-axis as $\Delta y_v$.

curvature $\rho$ as inputs is used for the computation of the environmental constraint:

$$\theta_{env} = \left(1 - \frac{M_v v_x^2}{2(l_f + l_r)^2} \frac{l_f C_f - l_r C_r}{C_f C_r}\right)(l_f + l_r)i_o\rho \qquad (14)$$

where $M_v$ is the mass of the vehicle, $l_f$ and $l_r$ are the distance from the gravity center to the front and rear axles, respectively, $C_f$ and $C_r$ are the fronts and rear cornering stiffness, and $i_o$ is the overall gear ratio from the steering angle to the tire angle.

A driver intent estimator is introduced to generate an approximation of the manual deviation away from the environmental constraint. It is assumed that a simple admittance model used to convert the driver torque to a desired future angle propagated by some time interval will provide a rough approximation of the driver intent[50]:

$$\theta_{int} = \int\int_t^{t+t_i} \frac{T_{tb}(t)}{J_{sw} + J_d} dt \qquad (15)$$

where $J_d$ is the driver inertia and $t_i$ is the propagation time. Both parameters can be tuned.

With the available measurements of the torque $T_{tb}$ and pinion angle $\theta_p$ as well as the estimate of the driver goal $\hat{\theta}_d$, the EKF[51] is developed for estimating the driver impedance[52]. The measurement of the pinion angle allows the decoupling of the steering wheel inertia from the dynamics of the pinion[53]. Consequently, Eq. (2), Eq. (3), Eq. (5), and Eq. (7) are discretized at time interval $\Delta t$ to form the plant model for the estimation.

$$x_{t+1} = f_t(x_t) + w_t \qquad (16)$$

$$y_t = h_t(x_t) + v_t \qquad (17)$$

where,

$$f_t = \begin{bmatrix} \theta_{sw,t} + \dot{\theta}_{sw,t}\Delta t \\ \dot{\theta}_{sw,t} + J_{sw}^{-1}(T_{d,t} - T_{tb,t})\Delta t \\ \hat{\theta}_{d,t} + \dot{\hat{\theta}}_{d,t}\Delta t \\ \dot{\hat{\theta}}_{d,t} \\ Z_{d,1,t} + T_{z,d}^{-1}(-Z_{d,1,t} + Z_{d,1,t-1})\Delta t \\ Z_{d,2,t} + T_{z,d}^{-1}(-Z_{d,2,t} + Z_{d,2,t-1})\Delta t \end{bmatrix}$$

$$h_t = \begin{bmatrix} T_{tb,t} \\ \hat{\theta}_{d,t} \\ \dot{\hat{\theta}}_{d,t} \end{bmatrix}$$

$$x_t = \begin{bmatrix} \theta_{sw,t} & \dot{\theta}_{sw,t} & \hat{\theta}_{d,t} & \dot{\hat{\theta}}_{d,t} & Z_{d,1,t} & Z_{d,2,t} \end{bmatrix}'$$

The following observer is formulated for the estimation of the driver impedance.

$$\hat{x}_{t+1/t} = f_t(\hat{x}_{t/t}) \qquad (18)$$

$$\hat{x}_{t/t} = \hat{x}_{t/t-1} + K_t(y_t - h_t(\hat{x}_{t/t-1})) \qquad (19)$$

The EKF gain is calculated as:

$$K_t = P_{t/t-1}\hat{H}_t'(\hat{H}_t P_{t/t-1}\hat{H}_t' + R)$$

where $P$ can be obtained by solving the Riccati equations:

$$P_{t+1/t} = \hat{F}_t P_{t/t}\hat{F}_t' + Q$$

$$P_{t/t} = P_{t/t-1} - P_{t/t-1}H_t'(\hat{H}_t P_{t/t-1}\hat{H}_t' + R)^{-1}\hat{H}_t P_{t/t-1}$$

where $\hat{x}$ is the state estimated by the EKF and $\hat{F}$ and $\hat{H}$ are Jacobian matrices, defined as follows.

$$\hat{F}_t = \left(\frac{\partial f_t(x_t)}{\partial x_t}\right)_{x_t = \hat{x}_t}, \quad \hat{H}_t = \left(\frac{\partial h_t(x_t)}{\partial x_t}\right)_{x_t = \hat{x}_t}$$

where $Q$ and $R$ are the covariance matrices of the process noise $w$ and observation noise $v$ respectively, which have to be tuned based on the modeling error and the noise level of the target system. Through computation of the prediction and correction[33] with Eq. (18) and Eq. (19), the last two components of $\hat{x}$ are estimated as the driver impedance $Z_d$.

**Inclusion of driver intent into the trajectory planning**. The arbitration rule allocates the control authority of the automation according to the preselected type of interaction. Manual intervention causes a deviation from the AD trajectory. Sustained input from the driver results in a steady interaction torque, and when released, the vehicle returns to the AD trajectory. This section presents the inclusion of driver intervention into trajectory planning to realize collaborative steering. For example, during a manually triggered lane change maneuver, it is necessary to reflect the driver intent in the trajectory planning. Hence, the reaction torque remains bounded along the maneuver and the driver does not have to apply a sustained torque to keep the vehicle in the new lane. These effects on the reaction torque represent haptic cues that communicate to the driver how the automated steering collaborated during the maneuver.

The proposed approach is inspired by the literature[27,28] for the integration of the driver intent into trajectory planning. However, rather than using the driver torque for the trajectory planning because of the absence of interactive steering control, the proposed approach uses the angular deviation resulting from the interaction. Consequently, collaborative steering is available only when the type of interaction enables a manual deviation from the AD trajectory, such as co-activity and collaboration. In the following, only the differences from the literature[27,28] are presented.

Inclusion consists in adding a term that represents the manual intent into the trajectory planning. At first, the yaw rate of the vehicle $\gamma_m$ caused by the manual intervention is computed from a single track vehicle model[54] with the driver angle $\theta_m$ as input (Fig. 10a):

$$\dot{x}_v = A_v x_v + B_v u_v$$
$$x_v = \begin{bmatrix} \beta \\ \gamma_m \end{bmatrix} u_v = \delta_m = \frac{\theta_m}{i_o}$$
$$A_v = \begin{bmatrix} a_{11} & a_{12} \\ a_{21} & a_{22} \end{bmatrix}, B_v = \begin{bmatrix} b_{11} \\ b_{21} \end{bmatrix}$$
$$a_{11} = \frac{-2(C_r + C_f)}{M_v v_x}, a_{12} = \frac{2(l_r C_r - l_f C_f)}{M_v v_x^2} - 1, \qquad (20)$$
$$a_{21} = \frac{2(l_r C_r - l_f C_f)}{I_z}, a_{22} = \frac{-2(l_r^2 C_r + l_f^2 C_f)}{I_z v_x},$$
$$b_{11} = \frac{2C_f}{M_v v_x}, b_{21} = \frac{2l_f C_f}{I_z}$$

where $\beta$ is the side slip angle, $v_x$ is the longitudinal velocity, and $I_z$ is the yaw moment of inertia of the vehicle. Second, a constant turn ratio and velocity (CTRV) model is used for converting the calculated yaw rate into a driver desired lateral deviation. The CTRV model enables the computation of the lateral deviation $\Delta y_d$ when the vehicle moves forward during a time horizon $t_s$ in stationary condition with constant vehicle longitudinal velocity $v_x$ and yaw rate $\gamma_m$. The kinematics are given as follows[55]:

$$\Delta y_d = \Delta y_v + \frac{v_x}{\gamma_m}(1 - \cos(t_s\gamma_m)) \qquad (21)$$

where $\Delta y_d$ and $\Delta y_v$ represent the lateral error between the driver desired lateral position and the AD trajectory and that between the current vehicle position and the AD trajectory as illustrated in Fig. 10b. The inclusion of the driver intent uses this estimate of the lateral deviation as a new corrective term into the trajectory planning. The cost function used to select the optimal lateral trajectory $y_{r,opt}$ from a predefined set of candidates $y_r(i,k)$ is augmented with the new corrective term.

$$C_y(i,k) = k_j J_y(i,k) + k_t t_f(k)$$
$$+ k_a(y_{rf}(i))^2 + k_m(y_{rf}(i) - \Delta y_d)^2 \qquad (22)$$

where $k_j, k_t, k_a, k_m \in \mathbb{R}$ are the weights of the cost function components. $k_j J_y(i,k)$ is the jerk-related term to account for driving comfort. $k_t t_f(k)$ is the time-related term. $k_a(y_{rf}(i))^2$ and $k_m(y_{rf}(i) - \Delta y_d)^2$ account for the deviation errors from both agents. The final lateral position $y_{rf}(i)$ is used with the completion time $t_f(k)$ for the computation of the trajectory candidates $y_r(i,k)$.

Notice that the selected optimal lateral trajectory is tracked during no driver intervention only. In the case of manual intervention, the optimal lateral trajectory is continuously computed at a frequency higher than the completion time $t_f$.

Inclusion of the driver intent into the trajectory planning is realized with the term of the lateral position error from the driver in the cost function (Eq. (22)). Consequently, the deviation caused by the driver intervention is propagated to the trajectory planning, thus preventing the occurrence of excessive and sustained interaction torque. Moreover, this assimilation transfers the manual correction of the AD trajectory consistently to the other actuators of the vehicle, such as the brakes and the accelerator (Fig. 1c).

**List of KPIs**. The KPIs used for the driver quantitative study are listed as follows:

- *Driver effort (DrE)*

Driver torque steering effort during the time of manoeuver:

$$DrE = \int_0^{t_{sc}} T_{tb}^2 dt \qquad (23)$$

- *Steering entropy (StE)* Algorithm to calculate the entropy:

1. Obtain the time-series steering angle data for each sampling time $dt$ ($dt$ was set to 150 ms in this study with reference to the literature[56]).
2. The future steering angle is predicted by quadratic Taylor expansion from the past three data points of the steering angle, and the prediction error between the predicted future steering angle and the actual steering angle is obtained.
3. Determine the 90 percentile value $\alpha$ centered at 0 degrees ($\alpha$ was set to 0.25 from the average prediction error distribution of all participants when driving in conventional manual mode).
4. Divide the frequency distribution of the prediction error into nine bins based on the range of $\alpha$ ($-5\alpha$, $-2.5\alpha$, $\alpha$, $-0.5\alpha$, $0.5\alpha$, $\alpha$, $2.5\alpha$, $5\alpha$).
5. Calculate StE from the proportion $P_i$ of each bin using the following formula:

$$StE = \sum_{i=1}^{9} P_i log_9 P_i \qquad (24)$$

## Data availability
The data that support the findings of this study are available from the corresponding authors upon reasonable request.

## Code availability
The code that supports the findings of this study is available from the corresponding authors upon reasonable request.

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

## Acknowledgements

This work is supported by JTEKT Corporation Research and Development Headquarters.

## Author contributions

Control concepts: T.N., R.F., and J.S.; Algorithm implementation: T.N.; Experiments: T.N.; Result analysis: T.N.; Manuscript writing: T.N., R.F., J.S., and H.B. All authors have read and edited the paper, and agree with its content.

## Ethics declaration

We have complied with all relevant ethical regulations and obtained informed consent from all participants. Guidelines for study procedures were provided by JTEKT Corporation.

## Competing interests

The authors declare no competing interests.
