## [Peer review file · Communications Engineering]

Reviewers' comments:

Reviewer #1 (Remarks to the Author):

Haptic Interaction Control and Trajectory Adaptation for Shared Steering
Tomohiro Nakade, Robert Fuchs, Hannes Bleuler, and Juerg Schiffmann

The authors provide a rather comprehensive review of the literature on shared control of steering and describe the need for an automation system that adapts to the driver without taking away control. Instead, the proposed automation enables manual intervention at any time (supporting operational-level collaboration) and adapts the path plan when appropriate (supporting tactical-level collaboration). The method section of the paper presents a realization of the proposed automation system, featuring an admittance-based control scheme compatible with standard electric power steering assist hardware. A study is conducted both in a driving simulator and in a vehicle on a multi-lane track to demonstrate the proposed driver/automation collaborative driving system. It is argued that the realization using an admittance-control system makes the proposed algorithms and hardware simple to implement in existing vehicle steering systems. The paper makes significant contributions, however some of the claims are overstated and important details are missing. In the following I will offer my suggestions for improving the paper.

An important component of the proposed method is the on-line estimation of driver impedance, using only readings of the pinion and steering wheel angles, the known road ("environment constraint") θ_{env} , and an estimation of the driver target θ_{int} . The driver target θ_{int} is in turn is estimated using the torque from the torsion bar on the steering column and a model of the driver biomechanics (inertia and "propagation time"). A Kalman Filter is used, following the methods introduced by ref [46]. The performance of this Kalman Filter at estimating the driver impedance parameters (stiffness and damping) is rather poor, as evident in Figures 10 and 11. The estimates carry a significant amount of persistent noise that appears to reflect the steering angle sinusoid (slalom maneuver). And the estimation of target angle θ_{int} and the impedance parameters are confounded. Indeed, the estimates do not converge to their true value, especially in the "Combined" estimation scheme used in Figure 11. A comment is offered in the narrative "Because of the abstract nature of the driver model, higher accuracy is of no particular value". This comment does not give the reader any real sense of trust in the ability of the Kalman Filter approach to generalize to maneuvers other than a constant amplitude sinusoid or maneuvers that include a non-zero value for the automation steering command θ_a . It appears that the estimation of impedance parameters would break down if the instructions to the driver (or the maneuver undertaken) were to change.

Further, the development of the Kalman Filter could be improved with a clear definition and distinction between the variables θ_d and $\hat{\theta}_d$. It is not clear whether the LHS of Eq (12) is the estimate maintained within the Kalman Filter (Eq. (17)) and not the state of the model (Eq. (16)) (which it seems would be introduced before the Kalman Filter is developed). There are many assumptions behind the driver model, drawn from references [44] (which is very old), ref [45], and the mechanical human model features shown in Figure 4. While in principle the simplicity of the human model may be at fault for the poor performance of the Kalman Filter at estimating impedance values, it is also quite possible that a minimal stability margin and higher frequency dynamics of the admittance display setup are the root of the cause for the poor convergence of the Kalman Filter. It was mentioned that stability was an issue in running the experiments and the controller had to be hand-tuned (see Limitations paragraphs in section IV.) I would recommend that the authors not try to overstate the performance of the realization of their proposed automation system. And anything the authors could do to more fully explain the operation and tuning of the Kalman Filter would be appreciated (and tuning of the closed loop control system as well).

The paper is somewhat difficult to follow, especially on the first reading. A second reading is necessary to piece together the components and contributions of the paper. The writing would profit if more signposts were placed for the reader, describing the aims of the authors and the logic of the order in

which topics are presented. For example, only in the last paragraph of the introduction does it become clear that the authors intend to combine many previously proposed methods for haptic shared control under previously proposed frameworks (operational/tactical) and to do so on admittance-display hardware. What preceded this statement of objective was a broad outline of mostly human factors concerns and motivations. Yet the aim of the paper and the contribution does not lie in the field of human factors, it lies primarily in technology development and implementation. Similarly, the Methodology section does not become transparent until the Results are presented, where the arbitration rules are tested in certain contexts. In particular the trajectory adaptation and arbitration system algorithms only became clear upon reading the Results. Another idea would be to include more cartoon graphs (expected results/sample graphs) to help the reader.

The development of Eq. (8) from Eqs. (7) and (6) is not clear. In the end, Eq. (14) also has an obscure relationship to Eq. (8) and Figure 4. These equations are central to the model and should be clarified.

In comparison to the development of the model of the human and their intended driving angles, the development of the trajectory planning is quite long and less interesting. Perhaps the literature can be referenced in this regard. Also in passing, the various trajectories in Figure 8 are "quantized" to discrete values of ΔT with the index k . In the end it is not clear that the driver would execute lane changes in this manner and the authors could make a comment in this regard.

At one point it is stated: "an automated vehicle which relies typically on sight only is augmented with the sense of touch." I am not sure if this is an accurate statement. It should be "... augmented with the driver intent/haptic interaction between driver and automation".

It would be nice to expand section II.F. Just a bit so that other researchers could reproduce the setup in Figures 9(a) and (b).

It would be worthwhile to mention in the narrative to Fig. 12 that the torque is lower in the collaboration phase resulting in lower conflicts.

The authors write: "However, it can be seen that as the automation impedance vanishes, the driver torque is not reacted, which tends to increase the automation impedance. This limit cycle...". What does the "limit cycle" mean? I think driver torque depends on automation torque, but does it only depend on that? According to Eq. (10), there are other components to it.

The authors also write: "Because the driver and automation impedances are complementary in collaboration mode, the torque peak observed is lower than that during co-activity (Fig. 13(b))." These features are not apparent from the figures. Just as in Fig. 13, the authors should add "total effort" on the Torque plots in Fig. 14 and 15.

Time and again the authors say things like "[with their method] The driver feels that his intervention has been assimilated resulting in intuitive collaborative drive." and "[with their method] intuitive haptic communication is made available to the driver", but these statements seem too bold considering the data only involves one human driver. The method seems very thoughtfully developed, promising, and effective, but a human subject experiment is clearly in order to actually show that this "multi-level haptic control" method is actually effective and intuitive. The authors need to acknowledge this. A comparison with existing haptic shared control methods is essential. It is important to establish that "multi-level haptic control" is more acceptable and intuitive than conventional methods especially without sacrificing/completely elimination the haptic cues from automation and road that are essential to ensure continuous interaction and feedback. It is not necessary to undertake all the human factors studies needed to establish these outcomes in this paper. However, it is necessary that these human factors studies remain future work and further, these studies would need to include more interesting driving scenarios than simply lane changes.

And a few smaller issues:

It was mentioned that "Smaller values of k enable stronger resistance and ultimately full rejection of the driver intervention." Could the authors clarify that smaller means more negative?

How is J_d in Eq. (14) defined or identified?

In Section III. A. what filter was used? What were the parameters of the filter design?

In Fig. 12, 14, 15. I think $L_{d,1}$ and $L_{d,2}$ should have hats on them. They are estimates.

The term "local deformation" should be revised. I did not understand it until section II.E. Deviation is a better term than deformation.

On pg 7 second para, typo. planning instead of planing.

It would be nice if the authors adopted the more conventional symbol "Z" for impedance and reserved "L" for observer or Kalman Filter gain.

The term "degree-of-freedom" is used several times, sometimes with its meaning from mechanics (number of independent generalized coordinate derivatives) and sometimes intending its meaning from the field of control (two transfer functions to be designed in either a feed-forward/feedback or inner/outer loop) and sometimes with its meaning in robotics (number of independent configuration variables).

The variable θ_m was used but then dropped and replaced (I believe) with θ_d or θ_{int} (not clear).

"Indeed these interaction types are those experienced..." Please de-reference "these". It is not clear which interaction types are being referenced.

The following statement could be expanded to better explain the operation of the Kalman Filter:

"Through an iterative calculation of the estimation and correction..."

The $\cos()$ term in Eq. (22) needs a reference or explanation.

A reference to the literature using measures of human input with Model Predictive Controllers would be useful to compare to the proposed approach. See for example:

Ercan, Ziya, et al. "Modeling, identification, and predictive control of a driver steering assistance system." IEEE Transactions on Human-Machine Systems 47.5 (2017): 700-710.

Shared steering control using safe envelopes for obstacle avoidance and vehicle stability, SM Erlen, S Fujita, JC Gerdes, IEEE Transactions on Intelligent Transportation Systems 17 (2), 441-451

Reference to the classical "bicycle model" of a four-wheeled vehicle would be helpful in connection with Eq. (21).

"when the driver impedance looses" should be something like "when the impedance of the driver diminishes".

"The originality of the proposed implementation summarizes as follows" should read instead: "The originality of the proposed implementation is summarized as follows".

Annotations in Figures 13-15 would be helpful, to indicate the timing of the lane changes

Reviewer #2 (Remarks to the Author):

The paper addresses an important problem of trajectory arbitration for a haptic shared control problem. Here are my comments to improve the readability of the paper.

1- Please label some of the important variables such as θ_a , θ_p and θ_d , .. in Figure 1.

2- What is T_{mot} in Eq 6 and how it is a function of θ_a .

3- It is not clear for me if the author assumes θ_a is known or it varies as L_a changes.

4- If the arbitration only happens by adapting L_a , then what is the main difference between this work and the work presented in the following paper "Modulation of Control Authority in Adaptive Haptic Shared Control Paradigms".

5 Figures in the results sections are very clean. I am wondering if there was any uncertainty in the system that can be captured with 95% confidence intervals in these Figures.

6-

Reviewer #3 (Remarks to the Author):

The work presented in this publication might constitute an interesting contribution to the field, but definitely not in its current form. I would argue for a reject in its current form., although I think the work might be a good first step towards a solid contribution. My main issues with this current manuscript include:

- a) the novelty with respect to the literature and commercial state-of-the-art is not sufficiently clear - and is overstated in the text;
- b) the methodology for the four evaluation experiments is not reported, rendering the experiments irreproducible;
- c) the assumptions and estimations around driver behavior are not validated, making the approach dubious at best
- d) the validation experiments are too limited and not grounded in human factors, also including only one participant, making it very difficult to generalize the findings towards the envisioned real-world use
- e) the conclusions are not clearly stated, and not discussed with respect to literature

0. NOVELTY

It seems the main contribution is to propose a trajectory-adapting shared control in real vehicles, due to the claim there is "sparse practical consideration on the nonlinear characteristics of a steering system and on the control stability", and the stated goal to "address this missing haptic link in automated driving within the limitation of mass produced hardware". The need and novelty is still not clear: many haptic lanekeeping assist systems are on the market, mass-produced, based on the two decades of research and innovation in this theme. Concept from literature have been implemented in real vehicles and tested successfully.

Also, trajectory adaptation has also been proposed by several authors already, and it is unclear how the proposed work compares to them. The work of Benloucif is acknowledged, but the novelty of the proposed approach is not argued well, nor are results compared to their work. Also earlier work (not cited by the authors) yielded very similar functionality for lanechanging, but without a need for all the intricate estimations of admittance and intent, for example:

Tsoi, K. K., Mulder, M., & Abbink, D. A. (2010, October). Balancing safety and support: Changing lanes with a haptic lane-keeping support system. In 2010 IEEE international conference on systems, man and cybernetics (pp. 1236-1243). IEEE.

Heil, T., Lange, A., & Cramer, S. (2016, November). Adaptive and efficient lane change path planning for automated vehicles. In 2016 IEEE 19th International Conference on Intelligent Transportation Systems (ITSC) (pp. 479-484). IEEE.

Or is the novelty that the "proof of concept of this multi-level haptic control represents a major breakthrough"? Benloucif et al proposed such multi-level haptics in his papers too, similarly Flemisch et al. (in more conceptual terms, granted). This leaves the real contribution hard to understand (and I would refrain from self-assessing your work as a major breakthrough).

1. INTRODUCTION

The introduction needs a very thorough revision.

First, the introduction is quite long and meandering, discussing important concepts (e.g. from the field of haptic shared control, human-automation interaction, human-human physical cooperation, physical HRI) without clarifying how these concepts are linked, and what the authors view is lacking from the

current state-of-the-art in shared control research.

The main problem the authors want to address, and their main contribution is very hard to find, and not well-defined when it finally comes at the end of the introduction. Part of it is mentioned in the conclusion (instead of there providing real conclusions.. see below).

Several times, I wondered if the authors fully grasped concepts from literature:

For example: adaptive automation is a concept not from the group of Burdet (cited as ref [14]), but from authors like Kaber, Inagaki, Parasuraman, who thought deeply and published much about it. The reference to Burdet's work is odd, I suggest the authors to read and use:

- Kaber, D. B., Riley, J. M., Tan, K. W., & Endsley, M. R. (2001). On the design of adaptive automation for complex systems. *International Journal of Cognitive Ergonomics*, 5(1), 37-57.

- Parasuraman, R., Bahri, T., Deaton, J. E., Morrison, J. G., & Barnes, M. (1992). Theory and design of adaptive automation in aviation systems. Catholic Univ of America Washington DC cognitive science lab.

- Byrne, E. A., & Parasuraman, R. (1996). Psychophysiology and adaptive automation. *Biological psychology*, 42(3), 249-268.

Only after reading the paper in full, did I start to understand why Jaresse's framework was explained in the introduction: the authors use this framework to define and estimate driver-automation roles. But the need to do so should be motivated in the introduction.

A second time I wondered about the grasp on literature, is when using the term blended control as an alternative to admittance control. Admittance control in robotics is used to render a virtual mechanical model, requiring a torque sensor. But what is the controlled system here: the steering wheel? The whole steering system? Also the authors are wrong in their statement that for haptic shared control "The inherent cost is that one agent performance is traded for the other." As demonstrated in several publications, drivers can amplify the torques from shared control, or resist them. There's no trading of performance, in those cases. Also "most vehicles on the market use the so-called traded control which is a simplification of blended control enabling manual or automated mode but not both together." How is traded control a simplification of blended control? This is very difficult to understand, and not defined well.

The figure 1 does little to help clarify this classification: what is the plant? What are the signals depicted by the arrows? As shown now, Figure 1a could depict both haptic shared control (if the agent outputs are torque) or input-mixing / blending shared control (if the agent outputs are position, which is then mixed inside the plant). Similarly for admittance control scheme is very vague, and misses detail.

The comparison between traded and shared control is elaborately made, as almost as if the authors are the first to propose the idea of shared control based on force feedback/haptic feedback. The general argumentation around haptic shared control proposed in the introduction has been made several times in more detail, including some of the works in the reference list, (Steele & Gillespie 2001; Abbink et al 2012) which are surprisingly hardly used to build the proposed novelty on, but also some other papers that provide much of the initial argumentation:

- Inagaki, T. (2003). Adaptive automation: Sharing and trading of control. *Handbook of cognitive task design*, 8, 147-169.

- Abbink, D. A., Carlson, T., Mulder, M., De Winter, J. C., Aminravan, F., Gibo, T. L., & Boer, E. R. (2018). A topology of shared control systems—finding common ground in diversity. *IEEE Transactions on Human-Machine Systems*, 48(5), 509-525.

I believe that if the authors describe their main innovation (see first point about novelty) much more quickly, and not argue again for the use of haptic shared control in general, the introduction would be shorter and more to the point.

2. METHODS

Estimation of stiffness and damping of the arms on the steering wheel is not trivial: time-variant, non-linear, and quite likely to be wrongly estimated (as can be seen in Figure 10 already, in case stiffness and damping are set a priori). The authors use an extended Kalman filtering approach, which has its own limitations and drawbacks. But since there is no clear methodology presented to validate the estimation of driver stiffness and damping (only a validation based on a priori settings of stiffness and damping, in experiment 1), the proposed method is not convincing. Where does the estimation work well enough, and where doesn't it? What is good enough? What is the impact of wrong estimations, on the driver control and experience? These are very important questions that are not addressed.

Much literature has shown how to estimate stiffness, damping and inertia (e.g. through frequency domain methods, or validation with electromyography or grip strength). These estimations depend on hand orientation on the steering wheel, individual characteristics, neuromuscular setting (giving way to torques or resisting them) just to name a few. Some of this literature is present in the reference list, without describing how this links to the chosen admittance estimation. I would advise the authors to relate their limb stiffness/damping estimation based on extended Kalman filtering to:

- Pick, A. J., & Cole, D. J. (2007). Dynamic properties of a driver's arms holding a steering wheel. *Proceedings of the Institution of Mechanical Engineers, Part D: Journal of Automobile Engineering*, 221(12), 1475-1486.
- Pick, A. J., & Cole, D. J. (2007). Driver steering and muscle activity during a lane-change manoeuvre. *Vehicle system dynamics*, 45(9), 781-805.
- Abbink, D. A., Cleij, D., Mulder, M., & van Paassen, M. M. (2012, October). The importance of including knowledge of neuromuscular behaviour in haptic shared control. In *2012 IEEE International Conference on Systems, Man, and Cybernetics (SMC)* (pp. 3350-3355). IEEE.
- Katzourakis, D. I., Abbink, D. A., Velenis, E., Holweg, E., & Happee, R. (2013). Driver's arms' time-variant neuromuscular admittance during real car test-track driving. *IEEE Transactions on Instrumentation and Measurement*, 63(1), 221-230.
- Abbink, D. A., Mulder, M., & Van Paassen, M. M. (2011, October). Measurements of muscle use during steering wheel manipulation. In *2011 IEEE International Conference on Systems, Man, and Cybernetics* (pp. 1652-1657). IEEE.
- Pronker, A. J., Abbink, D. A., Van Paassen, M. M., & Mulder, M. (2020). Estimating an LPV model of driver neuromuscular admittance using grip force as scheduling variable. *IEEE Transactions on Human-Machine Systems*, 50(5), 454-464.

None of the four experiments contain clear methodology, that can be reproduced. What is the parameterization of the EPS? What are the instructions to the driver in experiment 3 and 4? What are the vehicle characteristics? At what speed did the driver drive?

The third and fourth experiment contain only one participant, it seems. Without a human factors study there is no conclusion possible that all the estimations and assumptions on driver behavior actually result in an implementation that can adapt to individual trajectories (one of the main claims).

Also, the HRI design philosophy underlying the current work seems to go against the concept of shared control:

"Smaller values of enables stronger resistance and ultimately full rejection of the driver intervention." This would mean the driver can't overrule the automation, for example if a competition mode is estimated incorrectly. This goes against what the authors mention in the introduction: "Manual interventions are consistently considered and included in the path planning in view of closing the haptic loop of this proposed human-centered driving automation system."

Similarly, I have issues with the current implementation: "where the driver is not holding the steering

wheel, the control authority is naturally returned to the automation." This stimulates overreliance, the exact issue the authors claim to solve with their contribution! If authority lies with the automation when hands are off the steering wheel, drivers will have no incentive to put their hands on the wheel. And as is well-known from e.g. Tesla implementations, this is not simply solved with a hands-off warning system, or driver observation measurements. A thorough human factors evaluation would expose such issues of the chosen implementation, but is missing.

3. RESULTS

The authors report "Stability issue has been observed for low damping of the automation" and recommend further analysis. However, any detail around this is missing.

The variability of the measurements with the human participant are not shown. If there is none, there are no convincing results.

The experiments do not address major human factors evaluation concerns like acceptance, behavioral adaptation, or even a clear reporting of other behavioral findings (speed, steering wheel reversals) or the impact of misestimations.

4. Contribution and limitation.

I would expect a much more thorough discussion with respect to related work.

Also, main limitations are missing: in estimating admittance, in estimating the role, in the chosen experimental validation etc.

5. CONCLUSIONS

The conclusions are very qualitative and not clear: "the local deformation of the trajectory is meaningfully propagated to the path generation". Please stick to the real conclusions for which your study provides evidence, and link to that evidence.

----STYLE/GRAMMAR/SPELLING----

The reference style is not consistent.

The captions of all the figures are very short, making the figures hard to read without going through all the text.

A list of parameters would be helpful for any reader.

Finally, the manuscript contains too many spelling and grammar errors for my liking. Some examples:

Over reliance = over-reliance

Out of the loop problem = out-of-the-loop problem

"looses" = loses

"the driver torque is not reacted"

The word meaningful is often used without clarifying what constitutes meaningful or meaningless.

Trustful is not a word

Odd phrasing without explanations are often used in the text, like:

- "inclusion" seems a vague way to describe reference trajectory adaptation

- "The implemented framework requires the interaction type between driver and automation to be set by a higher level controller based on endogenous and exogenous information. "

What does that mean?

The authors would like to thank the reviewers for their thorough work. The very constructive comments are highly appreciated. Most of the comments have been implemented and highlighted in the revised manuscript. Our answers to the reviewers are given below in blue.

Reviewer #1:

The authors provide a rather comprehensive review of the literature on shared control of steering and describe the need for an automation system that adapts to the driver without taking away control. Instead, the proposed automation enables manual intervention at any time (supporting operational-level collaboration) and adapts the path plan when appropriate (supporting tactical-level collaboration). The method section of the paper presents a realization of the proposed automation system, featuring an admittance-based control scheme compatible with standard electric power steering assist hardware. A study is conducted both in a driving simulator and in a vehicle on a multi-lane track to demonstrate the proposed driver/automation collaborative driving system. It is argued that the realization using an admittance-control system makes the proposed algorithms and hardware simple to implement in existing vehicle steering systems. The paper makes significant contributions, however some of the claims are overstated and important details are missing. In the following I will offer my suggestions for improving the paper.

An important component of the proposed method is the on-line estimation of driver impedance, using only readings of the pinion and steering wheel angles, the known road (“environment constraint”) θ_{env} , and an estimation of the driver target θ_{int} . The driver target θ_{int} is in turn is estimated using the torque from the torsion bar on the steering column and a model of the driver biomechanics (inertia and “propagation time”). A Kalman Filter is used, following the methods introduced by ref [46]. The performance of this Kalman Filter at estimating the driver impedance parameters (stiffness and damping) is rather poor, as evident in Figures 10 and 11. The estimates carry a significant amount of persistent noise that appears to reflect the steering angle sinusoid (slalom maneuver). And the estimation of target angle θ_{int} and the impedance parameters are confounded. Indeed, the estimates do not converge to their true value, especially in the “Combined” estimation

scheme used in Figure 11. A comment is offered in the narrative “Because of the abstract nature of the driver model, higher accuracy is of no particular value”. This comment does not give the reader any real sense of trust in the ability of the Kalman Filter approach to generalize to maneuvers other than a constant amplitude sinusoid or maneuvers that include a non-zero value for the automation steering command θ_a . It appears that the estimation of impedance parameters would break down if the instructions to the driver (or the maneuver undertaken) were to change. Further, the development of the Kalman Filter could be improved with a clear definition and distinction between the variables θ_d and $\hat{\theta}_d$. It is not clear whether the LHS of Eq (12) is the estimate maintained within the Kalman Filter (Eq. (17)) and not the state of the model (Eq. (16)) (which it seems would be introduced before the Kalman Filter is developed). There are many assumptions behind the driver model, drawn from references [44] (which is very old), ref [45], and the mechanical human model features shown in Figure 4. While in principle the simplicity of the human model may be at fault for the poor performance of the Kalman Filter at estimating impedance values, it is also quite possible that a minimal stability margin and higher frequency dynamics of the admittance display setup are the root of the cause for the poor convergence of the Kalman Filter. It was mentioned that stability was an issue in running the experiments and the controller had to be hand-tuned (see Limitations paragraphs in section IV.) I would recommend that the authors not try to overstate the performance of the realization of their proposed automation system. And anything the authors could do to more fully explain the operation and tuning of the Kalman Filter would be appreciated (and tuning of the closed loop control system as well).

We followed the reviewer’s comments and modified the text on the estimation performance by the Kalman filter. Further, the manuscript was revised thoroughly to avoid overstatement. The model in the Kalman filter was modified to reduce the oscillatory behavior in estimates and an explanation was added with regards to the reasons why the estimates do not fully converge to the set value:

1. The authors agree, the performance of the Kalman filter to estimate the driver impedance is poor indeed. However, even with additional sensors, the driver goal and impedance are abstract concepts, which are very difficult, if not impossible, to measure accurately. The manuscript has been revised to emphasize this point. However, we also would like to point out, that the objective of the impedance estimation in the framework presented in this manuscript is not necessarily an

accurate value, but rather to realize the concept of the arbitration strategy. As the presented results suggest, even the inaccurate estimates are good enough for the implemented framework.

2. Further, despite improvements of the model in the Kalman filter, the oscillatory behavior in the estimates persists. This stems from two factors: the first one is that the estimated values deviate from the true values at the points where the driver torque and angle are zero. This is due to a singularity in the impedance calculation model used in the Kalman filter. In fact, the point at which the steering angle becomes zero and the timing of noise generation are synchronized. This trend is well-agreed with the literature [34]. The second factor is due to the rough estimation of the driver goal. Figure 11 shows that when the driver goal is underestimated, the driver impedance is overestimated and vice versa. This is evident from the relation between driver goal and impedance. Furthermore, as the estimation of the driver goal deviates from the true value, there is a misalignment between the model in the Kalman filter and the observed model, which tends to amplify the estimation noise occurring at the singularity. This is also evident from the previous literature [43].

The paper is somewhat difficult to follow, especially on the first reading. A second reading is necessary to piece together the components and contributions of the paper. The writing would profit if more signposts were placed for the reader, describing the aims of the authors and the logic of the order in which topics are presented. For example, only in the last paragraph of the introduction does it become clear that the authors intend to combine many previously proposed methods for haptic shared control under previously proposed frameworks (operational/tactical) and to do so on admittance-display hardware. What preceded this statement of objective was a broad outline of mostly human factors concerns and motivations. Yet the aim of the paper and the contribution does not lie in the field of human factors, it lies primarily in technology development and implementation. Similarly, the Methodology section does not become transparent until the Results are presented, where the arbitration rules are tested in certain contexts. In particular the trajectory adaptation and arbitration system algorithms only became clear upon reading the Results. Another idea would be to include more cartoon graphs (expected results/sample graphs) to help the reader.

We followed the reviewer's suggestion and have made significant revisions to the

introduction section to make it more concise and to the point, which makes the paper easier to understand on first reading.

We have now clearly defined our proposed multi-level haptic shared control framework, which includes the interaction, arbitration, and inclusion in the objective statement in the introduction. Furthermore, we have improved the explanations in the captions of the Fig. 2 for improving the visual support as well.

The development of Eq. (8) from Eqs. (7) and (6) is not clear. In the end, Eq. (14) also has an obscure relationship to Eq. (8) and Figure 4. These equations are central to the model and should be clarified.

The development of these equations has been made more transparent in the reviewed version.

In comparison to the development of the model of the human and their intended driving angles, the development of the trajectory planning is quite long and less interesting. Perhaps the literature can be referenced in this regard.

We followed the reviewer's suggestion and changed the manuscript by limiting the description to the differences from the cited literature.

Also in passing, the various trajectories in Figure 8 are "quantized" to discrete values of ΔT with the index k . In the end it is not clear that the driver would execute lane changes in this manner and the authors could make a comment in this regard.

The completion time of the candidate's trajectory and the update timing of the candidate's path is different. This means that the shape of the updated trajectory is not limited to the candidate's trajectory and can be modified during the intervention. This point is made clearer in the revised manuscript.

At one point it is stated: "an automated vehicle which relies typically on sight only is augmented with the sense of touch." I am not sure if this is an accurate statement. It should be "... augmented with the driver intent/haptic interaction between driver and automation".

Corrected.

It would be nice to expand section II.F. Just a bit so that other researchers could reproduce the setup in Figures 9(a) and (b).

The requirement for the driver has been added, so that the presented data can be reproduced.

It would be worthwhile to mention in the narrative to Fig. 12 that the torque is lower in the collaboration phase resulting in lower conflicts.

A section about a quantitative study over several drivers (III-E) has been added, which addresses the reviewer's concerns.

The authors write: "However, it can be seen that as the automation impedance vanishes, the driver torque is not reacted, which tends to increase the automation impedance. This limit cycle... ". What does the "limit cycle" mean? I think driver torque depends on automation torque, but does it only depend on that? According to Eq. (10), there are other components to it.

Originally, we wanted to discuss the feedback loop between the driver and the automation impedance variation. However, since this is not the main focus of the manuscript, we have decided to remove this point to avoid confusion.

The authors also write: "Because the driver and automation impedances are complementary in collaboration mode, the torque peak observed is lower than that during co-activity (Fig. 13(b))." These features are not apparent from the figures. Just as in Fig. 13, the authors should add "total effort" on the Torque plots in Fig. 14 and 15.

A section about the quantitative study over several drivers (III-E) has been added, which addresses the reviewer's concerns.

Time and again the authors say things like "[with their method] The driver feels that his intervention has been assimilated resulting in intuitive collaborative drive." and "[with their method] intuitive haptic communication is made available to the driver", but these statements seem too bold considering the data only involves one human driver. The method seems very thoughtfully developed, promising, and effective, but a human

subject experiment is clearly in order to actually show that this "multi-level haptic control" method is actually effective and intuitive. The authors need to acknowledge this. A comparison with existing haptic shared control methods is essential. It is important to establish that "multi-level haptic control" is more acceptable and intuitive than conventional methods especially without sacrificing/completely elimination the haptic cues from automation and road that are essential to ensure continuous interaction and feedback. It is not necessary to undertake all the human factors studies needed to establish these outcomes in this paper. However, it is necessary that these human factors studies remain future work and further, these studies would need to include more interesting driving scenarios than simply lane changes.

The reviewer raises a very valid point. In order to address this, a section on the evaluation of the human factor has been added to maintain our main statement (III-E). Through five participants and the evaluation of two criteria (driver effort and steering entropy), this evaluation suggests that the proposed control framework has the potential to achieve smooth maneuvers with less effort for a wide variety of drivers.

And a few smaller issues:

It was mentioned that "Smaller values of k enable stronger resistance and ultimately full rejection of the driver intervention." Could the authors clarify that smaller means more negative?

Yes, we intended that smaller means more negative. Since this seems obvious, we did not modify the related statement.

How is J_d in Eq. (14) defined or identified?

Tuning guidelines have been added to the manuscript and the values used for implementation have been added to the parameters in Table 1.

In Section III. A.m what filter was used? What were the parameters of the filter design?

The use of a low-pass filter has been removed since the oscillatory behavior could be improved by modifying the Kalman filter. The section has been adapted accordingly.

In Fig. 12, 14, 15. I think $L_{d,1}$ and $L_{d,2}$ should have hats on them. They are estimates.

Corrected.

The term "local deformation" should be revised. I did not understand it until section II.E. Deviation is a better term than deformation.

We uniformized the manuscript by using "deviation".

On pg 7 second para, typo. planning instead of planing.

Corrected.

It would be nice if the authors adopted the more conventional symbol "Z" for impedance and reserved "L" for observer or Kalman Filter gain.

Corrected.

The term "degree-of-freedom" is used several times, sometimes with its meaning from mechanics (number of independent generalized coordinate derivatives) and sometimes intending its meaning from the field of control (two transfer functions to be designed in either a feed-forward/feedback or inner/outer loop) and sometimes with its meaning in robotics (number of independent configuration variables).

We rephrased in more specific terms to avoid confusion.

The variable θ_m was used but then dropped and replaced (I believe) with θ_d or θ_{int} (not clear).

The definitions of θ_m and θ_d are different. θ_m is the deviation resulting from the arbitration, θ_d is the driver goal that corresponds to where driver intent to go. The explanation on that and Figure 4 have been added to illustrate some of the parameters.

"Indeed these interaction types are those experienced..." Please de-reference "these".

It is not clear which interaction types are being referenced.

Corrected.

The following statement could be expanded to better explain the operation of the Kalman Filter: "Through an iterative calculation of the estimation and correction...."

The "iterative" have been replaced simply by the "computation", and a related citation has been added.

The $\cos()$ term in Eq. (22) needs a reference or explanation.

A citation for the \cos term has been added. This term corresponds to geometric transformations, which are described in Fig. 7

A reference to the literature using measures of human input with Model Predictive Controllers would be useful to compare to the proposed approach. See for example: Ercan, Ziya, et al. "Modeling, identification, and predictive control of a driver steering assistance system." IEEE Transactions on Human-Machine Systems 47.5 (2017): 700-710.

Shared steering control using safe envelopes for obstacle avoidance and vehicle stability, SM Erlien, S Fujita, JC Gerdes, IEEE Transactions on Intelligent Transportation Systems 17 (2), 441-451

This is indeed an interesting reference paper that explains some part of the arbitration strategies. It has been added to the state-of-the-art section.

Reference to the classical "bicycle model" of a four-wheeled vehicle would be helpful in connection with Eq. (21).

The citation has been added.

"when the driver impedance loses" should be something like "when the impedance of the driver diminishes".

We replaced "looses" by "goes to zero".

"The originality of the proposed implementation summarizes as follows" should read instead: "The originality of the proposed implementation is summarized as follows".

Corrected.

Annotations in Figures 13-15 would be helpful, to indicate the timing of the lane changes

We have added the lane marking in Figures 13-15 so that the readers can easily understand the crossing point of the lane marking. The starting of lane change corresponds to the point, where the measured driver torque (T_{tb}) arises.

Reviewer #2:

The paper addresses an important problem of trajectory arbitration for a haptic shared control problem. Here are my comments to improve the readingness of the paper.

- 1- Please label some the important variables such as θ_a , θ_p and θ_d in Figure1.

All symbols, their description, and detail explanation have been added in Figure 1.

- 2- What is T_{mot} in Eq 6 and how it is function of θ_a .

T_{mot} is the final motor command to realize the admittance control framework. In order to explain the detail of T_{mot} , Figure 4 has been added to the reviewed version.

- 3- It is not clear for me if the author assumes θ_a is known or it varies as the changes in L_a .

No, θ_a is computed from the AD trajectory and the trajectory tracking block in Figure 2. L_a is a function of θ_a , θ_d , and L_d .

- 4- If the arbitration only happens by adapting L_a , then what is the main difference between this work and the work presented in the following paper "Modulation of Control Authority in Adaptive HapticShared Control Paradigms".

Our proposal is the pragmatic implementation of multi-level haptic shared control, which is proposed as a more abstract and less complete concept in the state-of-the-art. The citation the reviewer mentioned corresponds to the arbitration part we proposed. To avoid the confusion, we clearly redefined our multi-level haptic control framework and added an explanation that highlights the differences from literature.

- 5- Figures in the results sections are very clean. I am wondering if there was any uncertainty in the system that can be captured with 95% confidence intervals in these Figures.

There are no system uncertainties. The plot in Figures 9-14 correspond to the actual measurement. For the evaluation on the deviation for the drivers (human users), we added a new section on a "Driver quantitative study" (III-E).

Reviewer #3:

The work presented in this publication might constitute an interesting contribution to the field, but definitely not in its current form. I would argue for a reject in its current form., although I think the work might be a good first step towards a solid contribution. My main issues with this current manuscript include:

- a) the novelty with respect to the literature and commercial state-of-the-art is not sufficiently clear - and is overstated in the text;

The introduction has been significantly revised to clarify and emphasize on our novel contribution. The details are explained below.

- b) the methodology for the four evaluation experiments is not reported, rendering the experiments irreproducible;

The description of the experimental procedure has been extended and the relevant parameters used in this paper have been summarized in Table 3. In our opinion the experiments are now reproducible.

- c) the assumptions and estimations around driver behavior are not validated, making the approach dubious at best

Additional discussion on driver behavior have been added. Furthermore, a new section has been added to include experimental data gathered on the test vehicle driven by several drivers to assess the technical viability and performance of the proposed framework (III-E).

- d) the validation experiments are too limited and not grounded in human factors, also including only one participant, making it very difficult to generalize the findings towards the envisioned real-world use

Additional test results involving several drivers have been added to provide more experimental evidence for the viability of the proposed control framework for a wide variety of drivers (III-E).

- e) the conclusions are not clearly stated, and not discussed with respect to literature

A detailed discussion on the proposed control framework related to state-of-the-art has been added in the contributions and limitations section. Furthermore, the conclusion has been updated to make them more clear and to better highlight our contribution.

0. NOVELTY

It seems the main contribution is to propose a trajectory-adapting shared control in real vehicles, due to the claim there is “sparse practical consideration on the nonlinear characteristics of a steering system and on the control stability”, and the stated goal to

“address this missing haptic link in automated driving within the limitation of mass-produced hardware”. The need and novelty are still not clear: many haptic lane keeping assist systems are on the market, mass-produced, based on the two decades of research and innovation in this theme. Concept from literature have been implemented in real vehicles and tested successfully.

Also, trajectory adaptation has also been proposed by several authors already, and it is unclear how the proposed work compares to them. The work of Benloucif is acknowledged, but the novelty of the proposed approach is not argued well, nor are results compared to their work. Also, earlier work (not cited by the authors) yielded very similar functionality for lane changing, but without a need for all the intricate estimations of admittance and intent, for example:

Tsoi, K. K., Mulder, M., & Abbink, D. A. (2010, October). Balancing safety and support: Changing lanes with a haptic lane-keeping support system. In 2010 IEEE international conference on systems, man and cybernetics (pp. 1236-1243). IEEE.

Heil, T., Lange, A., & Cramer, S. (2016, November). Adaptive and efficient lane change path planning for automated vehicles. In 2016 IEEE 19th International Conference on Intelligent Transportation Systems (ITSC) (pp. 479-484). IEEE.

Or is the novelty that the “proof of concept of this multi-level haptic control represents a major breakthrough”? Benloucif et al proposed such multi-level haptics in his papers too, similarly Flemisch et al. (in more conceptual terms, granted). This leaves the real contribution hard to understand (and I would refrain from self-assessing your work as a major breakthrough).

Yes, all the concepts of admittance control, arbitration and inclusion have been presented previously, but independently. Here, however, the different concepts are merged together for the first time and tested for the first time in a real environment (i.e. automated driving with haptic feedback) under the constraint of no added sensors.

1. INTRODUCTION

The introduction needs a very thorough revision.

First, the introduction is quite long and meandering, discussing important concepts (e.g., from the field of haptic shared control, human-automation interaction, human-

human physical cooperation, physical HRI) without clarifying how these concepts are linked, and what the authors view is lacking from the current state-of-the-art in shared control research.

The main problem the authors want to address, and their main contribution is very hard to find, and not well-defined when it finally comes at the end of the introduction. Part of it is mentioned in the conclusion (instead of these providing real conclusions. see below).

Several times, I wondered if the authors fully grasped concepts from literature:

For example: adaptive automation is a concept not from the group of Burdet (cited as ref [14]), but from authors like Kaber, Inagaki, Parasuraman, who thought deeply and published much about it. The reference to Burdet's work is odd, I suggest the authors to read and use:

- Kaber, D. B., Riley, J. M., Tan, K. W., & Endsley, M. R. (2001). On the design of adaptive automation for complex systems. *International Journal of Cognitive Ergonomics*, 5(1), 37-57.

- Parasuraman, R., Bahri, T., Deaton, J. E., Morrison, J. G., & Barnes, M. (1992). Theory and design of adaptive automation in aviation systems. Catholic Univ of America Washington DC cognitive science lab.

- Byrne, E. A., & Parasuraman, R. (1996). Psychophysiology and adaptive automation. *Biological psychology*, 42(3), 249-268.

Only after reading the paper in full, did I start to understand why Jaresse's framework was explained in the introduction: the authors use this framework to define and estimate driver-automation roles. But the need to do so should be motivated in the introduction.

The authors agree with the reviewer and have completely rewritten the introduction. The objective statement has majorly been revised so that the overall configuration of the proposed control algorithm is clear at a glance. A statement has been added that clearly defines the negotiation between the automation and driver by means of an arbitration strategy, as well as the assimilation of the automation trajectory by means of inclusion assuming an interactive framework.

A second time I wondered about the grasp on literature, is when using the term blended control as an alternative to admittance control. Admittance control in robotics is used to render a virtual mechanical model, requiring a torque sensor. But what is the

controlled system here: the steering wheel? The whole steering system?

The “plant” was used to represent the entire steering system. Conventional steering systems typically have an angle sensor and a torque sensor installed. We used this torque sensor to measure the driver input to implement the admittance control framework. To avoid confusion, the definition of “plant” has been added.

Also, the authors are wrong in their statement that for haptic shared control “The inherent cost is that one agent performance is traded for the other.” As demonstrated in several publications, drivers can amplify the torques from shared control, or resist them. There’s no trading of performance, in those cases.

Here we do not agree with the reviewer’s comment. Blended control requires a lower gain in position control to allow driver intervention. Alternatively, the performance must be lowered by modulating a gain for the automation torque input. In blended control, driver intervention and the tracking performance of automation are always inversely related and are problems to be solved. We solved them with the interactive control based on admittance control.

Also “most vehicles on the market use the so-called traded control which is a simplification of blended control enabling manual or automated mode but not both together.” How is traded control a simplification of blended control? This is very difficult to understand, and not defined well.

We followed the reviewer’s suggestion and deleted these sentences.

The figure 1 does little to help clarify this classification: what is the plant? What are the signals depicted by the arrows? As shown now, Figure 1a could depict both haptic shared control (if the agent outputs are torque) or input-mixing / blending shared control (if the agent outputs are position, which is then mixed inside the plant). Similarly for admittance control scheme is very vague, and misses detail.

Vague expressions have been eliminated and more details provided. Especially, all the symbols and related explanations have been added in the caption of Figure 1 to make the difference between admittance control and blended control clearer.

The comparison between traded and shared control is elaborately made, as almost as if the authors are the first to propose the idea of shared control based on force feedback/haptic feedback. The general argumentation around haptic shared control proposed in the introduction has been made several times in more detail, including some of the works in the reference list, (Steele & Gillespie 2001; Abbink et al 2012) which are surprisingly hardly used to build the proposed novelty on, but also some other papers that provide much of the initial argumentation:

- Inagaki, T. (2003). Adaptive automation: Sharing and trading of control. Handbook of cognitive task design, 8, 147-169.

- Abbink, D. A., Carlson, T., Mulder, M., De Winter, J. C., Aminravan, F., Gibo, T. L., & Boer, E. R. (2018). A topology of shared control systems—finding common ground in diversity. IEEE Transactions on Human-Machine Systems, 48(5), 509-525.

I believe that if the authors describe their main innovation (see first point about novelty) much more quickly, and not argue again for the use of haptic shared control in general, the introduction would be shorter and more to the point.

The authors fully agree. We followed the reviewer's suggestion and removed the detailed description of the haptic shared control. Furthermore, we have revised the manuscript by focusing on the proposed multi-level haptic shared control (interaction, arbitration, and inclusion), which is the most important argument of this study.

2. METHODS

Estimation of stiffness and damping of the arms on the steering wheel is not trivial: time-variant, non-linear, and quite likely to be wrongly estimated (as can be seen in Figure 10 already, in case stiffness and damping are set a priori). The authors use an extended Kalman filtering approach, which has its own limitations and drawbacks. But since there is no clear methodology presented to validate the estimation of driver stiffness and damping (only a validation based on a priori settings of stiffness and damping, in experiment 1), the proposed method is not convincing. Where does the estimation work well enough, and where doesn't it? What is good enough? What is the impact of wrong estimations, on the driver control and experience? These are very important questions that are not addressed.

As the reviewer pointed out, the proposed estimation method has limitations, and the consistency of the estimates with the actual driver intention has not been validated. The driver goal and the impedance are vague concepts indeed, and even if additional sensors (i.e. EEG or EMG) were available, accurate estimation would be difficult, especially in a practical way. Unfortunately, with the current method it is not clear whether changes in driver torque are due to changes in driver goal or to changes impedance. Here, in order to avoid this observability issue, the driver impedance is simply assumed to be the result of fitting the driver goal calculated from the interaction torque. Under this limitation, we proposed a way to extract the variation of driver impedance so that arbitration can be implemented. The accurate estimation is certainly important, but to realize the multi-level haptic shared control, the driver impedance estimation is simplified and bonded to the practical approximation. We have changed the manuscript so that this assumption and the resulting limitations are clarified for the readers. Further, approaches have been described on how to tune the driver goal estimation and to ensure the arbitration strategy works in a safe manner.

Much literature has shown how to estimate stiffness, damping and inertia (e.g., through frequency domain methods, or validation with electromyography or grip strength). These estimations depend on hand orientation on the steering wheel, individual characteristics, neuromuscular setting (giving way to torques or resisting them) just to name a few. Some of this literature is present in the reference list, without describing how this links to the chosen admittance estimation. I would advise the authors to relate their limb stiffness/damping estimation based on extended Kalman filtering to:

- Pick, A. J., & Cole, D. J. (2007). Dynamic properties of a driver's arms holding a steering wheel. *Proceedings of the Institution of Mechanical Engineers, Part D: Journal of Automobile Engineering*, 221(12), 1475-1486.
- Pick, A. J., & Cole, D. J. (2007). Driver steering and muscle activity during a lane-change manoeuvre. *Vehicle system dynamics*, 45(9), 781-805.
- Abbink, D. A., Cleij, D., Mulder, M., & van Paassen, M. M. (2012, October). The importance of including knowledge of neuromuscular behaviour in haptic shared control. In *2012 IEEE International Conference on Systems, Man, and Cybernetics (SMC)* (pp. 3350-3355). IEEE.
- Katzourakis, D. I., Abbink, D. A., Velenis, E., Holweg, E., & Happee, R. (2013). Driver's arms' time-variant neuromuscular admittance during real car test-track driving. *IEEE*

Transactions on Instrumentation and Measurement, 63(1), 221-230.

- Abbink, D. A., Mulder, M., & Van Paassen, M. M. (2011, October). Measurements of muscle use during steering wheel manipulation. In 2011 IEEE International Conference on Systems, Man, and Cybernetics (pp. 1652-1657). IEEE.

- Pronker, A. J., Abbink, D. A., Van Paassen, M. M., & Mulder, M. (2020). Estimating an LPV model of driver neuromuscular admittance using grip force as scheduling variable. IEEE Transactions on Human-Machine Systems, 50(5), 454-464.

We followed the reviewer's suggestion and cited the references on the driver arm impedance estimation and their relation with our proposal have been described. First, the ideas of limb impedance estimation and control gain estimation proposed in our study are fundamentally different. The estimation of the limbs does not consider the fluctuation of the driver goal (target angle) and tries to fit all the fluctuations in the relationship between the torque and angle as the limb impedance. With this approach, it is not possible to extract the impedance variation, which is only used for the target tracking task. In this proposal, the driver goal and control impedance are roughly estimated only with available sensors under the assumption that the driver has a unique control target, and the impedance is varied to track this target. The difficulty of estimation is overwhelmingly different due to this assumption and limitation. As the number of the variables increases for the driver goal, the estimation performance is lower than that in the citations suggested by the reviewer. Certainly, this is a constraint on our control, but it is important as a concept, and improvement of estimation performance is an important future task. These points have been included and clarified in the manuscript.

None of the four experiments contain clear methodology, that can be reproduced. What is the parameterization of the EPS? What are the instructions to the driver in experiment 3 and 4? What are the vehicle characteristics? At what speed did the driver drive?

The instructions to the driver have been clearly described in the experimental configuration section. The parameters related to EPS and vehicle characteristics have been summarized in Tables 2 and 3. In our opinion the experiments can be reproduced with this information.

The third and fourth experiment contain only one participant, it seems. Without a

human factors study there is no conclusion possible that all the estimations and assumptions on driver behavior actually result in an implementation that can adapt to individual trajectories (one of the main claims).

For the human factor discussion, and following the reviewer comments, an additional quantitative study was carried out with five participants and the results added in a new section (III-E). As suggested by the resulting driver effort and steering entropy, the proposed multi-level haptic shared control framework has the potential to achieve smooth maneuvers with less effort for a wide range of drivers. Of course, statistically more relevant data (more samples) is needed to fully assess the potential of the proposed framework. Nonetheless the presented results are already exciting and very promising.

Also, the HRI design philosophy underlying the current work seems to go against the concept of shared control:

“ Smaller values of enables stronger resistance and ultimately full rejection of the driver intervention.” This would mean the driver can't overrule the automation, for example if a competition mode is estimated incorrectly.

The determination of the interaction type is outside the scope of this study since this would be the task of a higher ranked control framework in the vehicle. However, it is clearly a subject for future work. A discussion point on this limitation has been added in the limitation section.

This goes against what the authors mention in the introduction: “Manual interventions are consistently considered and included in the path planning in view of closing the haptic loop of this proposed human-centered driving automation system.” Similarly, I have issues with the current implementation: “where the driver is not holding the steering wheel, the control authority is naturally returned to the automation.” This stimulates overreliance, the exact issue the authors claim to solve with their contribution! If authority lies with the automation when hands are off the steering wheel, drivers will have no incentive to put their hands on the wheel. And as is well-known from e.g., Tesla implementations, this is not simply solved with a hands-off warning system, or driver observation measurements. A thorough human factors evaluation would expose such issues of the chosen implementation, but is missing.

The reviewer raises a delicate point indeed. The manuscript has been corrected to state that the results of the arbitration strategy are reflected in the trajectory adaptation. Further, we are aware of Tesla's out-of-loop problem.

Fuchs, R., Sakai, Y., Tamura, T. et al. The Rules of Engagement for Partially Automated Driving. *ATZ Worldwide* 124, 16–21 (2022). <https://doi.org/10.1007/s38311-022-0818-7>

This could be solved by using the collaboration mode and the cooperation (assist) mode appropriately, but this setting way of interaction type is not the main topic of this article. The proposed framework addresses the challenge how to control the steering actuator after the interaction type is selected. To avoid over-reliance and to ensure the misdetection of competition, the interaction type must be selected dynamically based on the traffic conditions or driver state through a higher-level control at the vehicle management level.

3. RESULTS

The authors report “Stability issue has been observed for low damping of the automation” and recommend further analysis. However, any detail around this is missing.

This statement has been removed as it is not the focus of this study.

The variability of the measurements with the human participant are not shown. If there is none, there are no convincing results.

The results of the subject tests were added (III-E), which address the reviewer’s comment.

The experiments do not address major human factors evaluation concerns like acceptance, behavioral adaptation, or even a clear reporting of other behavioral findings (speed, steering wheel reversals) or the impact of misestimations.

The human factor evaluation is not the main concern of this study. The main

contribution is the implementation of the multi-loop haptics control concept in real applications. The introduction has been revised to emphasize this claim.

4. Contribution and limitation.

I would expect a much more thorough discussion with respect to related work. Also, main limitations are missing: in estimating admittance, in estimating the role, in the chosen experimental validation etc.

The contribution and limitation section has been revised to include the estimation limitation, missing point of interaction type setting, and human subject evaluation. Together with the revised introduction the paper is now much more focused and concise.

5. CONCLUSIONS

The conclusions are very qualitative and not clear: “the local deformation of the trajectory is meaningfully propagated to the path generation”. Please stick to the real conclusions for which your study provides evidence, and link to that evidence.

The data of subject evaluation were linked to clarify the conclusions.

---STYLE/GRAMMAR/SPELLING---

The reference style is not consistent.

Corrected.

The captions of all the figures are very short, making the figures hard to read without going through all the text.

Detailed captions have been added to all the figures and tables.

A list of parameters would be helpful for any reader.

A complete list of symbols and parameters has been added.

Finally, the manuscript contains too many spelling and grammar errors for my liking. Some examples:

Over reliance = over-reliance

Out of the loop problem = out-of-the-loop problem

“looses” = loses

“the driver torque is not reacted”

The word meaningful is often used without clarifying what constitutes meaningful or meaningless.

Trustful is not a word

The manuscript has been corrected and checked thoroughly for spelling and grammar errors.

Odd phrasings without explanations are often used in the text, like:

- “inclusion” seems a vague way to describe reference trajectory adaptation
- “The implemented framework requires the interaction type between driver and automation to be set by a higher-level controller based on endogenous and exogenous information. “

What does that mean?

Definitions of inclusion, endogenous and exogenous information have been added.

Reviewers' comments:

Reviewer #2 (Remarks to the Author):

Thanks for addressing the raised issue. Very nice work.

Reviewer #3 (Remarks to the Author):

I am pleased to see the authors took the feedback from myself and the other reviewers seriously, thoroughly revising the introduction, improving the clarity of figures and blockschemes, running additional experiments, and sharpening the conclusions. There is still work to be done, in my opinion, to further clarify the motivation, the value with respect to other approaches in literature, and some of the analyses. I propose another revision, to address these comments.

INTRODUCTION

The introduction has been greatly improved, is much more legible and to the point. Having said that, I still have some concerns.

A small concern lies around the claims of what constitutes shared control. For example "Shared steering control is available under a preset driver torque threshold". This is not necessarily the case, there are also many examples where shared control is never switched off like that (actually, with sufficiently high torque tuning and such switches, shared control becomes traded control) . I would advise clearly defining shared control, and relating your definitions to those proposed in a relevant review paper of shared control, that proposes a clear definition around shared control: ---Abbink, D. A., Carlson, T., Mulder, M., De Winter, J. C., Aminravan, F., Gibo, T. L., & Boer, E. R. (2018). A topology of shared control systems—finding common ground in diversity. *IEEE Transactions on Human-Machine Systems*, 48(5), 509-525.

"Nature of the issue". I like this heading, it provides a clear anchor to the reader. However, when I read the section, I still was unclear about the real nature of the issue.

Why is there a need for the "development of a generic control framework consistent across all cognitive levels"? What cognitive levels do you mean, and what is the trouble of a framework that is "inconsistent" across these? Perhaps you hint here at what you later state more clearly (The integration of interaction, arbitration, and inclusion into a generic multi-level control framework represents the main contribution of this work.) If that is the case, please clarify this accordingly, or else clarify what you mean with "cognitive levels".

Second, the "major disadvantages" of what the author calls blended control are not completely clarified yet. These issues seem to revolve around

- lowered tracking performance, due to tuning of a PID controller (perhaps adding the PID controller in Fig 1 or 2 would help clarify the control-theoretic issue you address)
- driver comfort and acceptance
- a trade-off (as mentioned in the conclusions also), but without clarifying that trade-off, or the consequences of that trade-off

Central seems to be: "Ideally, perfect tracking is expected in the absence of driver input, while low rejection performance is required to enable manual intervention." But, there are examples of control architectures that do not suffer from this. For example

- the works of Benloucif et al that you cite,
- or architectures described in papers you don't cite (like van Paassen et al; Scholtens et al; see refs below) - that also include what you call "inclusion"

----Paassen et al. (2017). Four design choices for haptic shared control. In *Advances in Aviation Psychology*, Volume 2 (pp. 237-254)

----Scholtens, W. et al. (2018, October). A new haptic shared controller reducing steering conflicts. In *2018 IEEE International Conference on Systems, Man, and Cybernetics (SMC)* (pp. 2705-2710). IEEE.

- similarly, you currently acknowledge only a limited cross-section of studies that do trajectory adaptation. There are more studies that can both generate haptic control in real-vehicles as well as adapt trajectories (e.g. Kolekar et al, see ref below).

---Kolekar, S., Petermeijer, B., Boer, E., de Winter, J., & Abbink, D. (2021). A risk field-based metric correlates with driver's perceived risk in manual and automated driving: A test-track study. *Transportation research part C: emerging technologies*, 133, 103428.

There are also other ways to do haptic shared control that can adapt trajectories when (double) lanechanging, that do not require arbitration. how does your approach compare to that? See:

---Tsoi, K. K., Mulder, M., & Abbink, D. A. (2010). Balancing safety and support: Changing lanes with a haptic lane-keeping support system. In *2010 IEEE international conference on systems, man and cybernetics* (pp. 1236-1243). IEEE.

All in all, currently the need to combine all of the three elements (interaction/arbitration/inclusion) - and it's expected benefit compared to the literature, besides the combination in and of itself - does not follow logically from the introduction still, nor why the Kalman Filtering is necessary (constituting such a large part of the paper). At least I would prefer an even stronger motivation why these three are needed, and what the added value is compared to literature (e.g., the works of Benloucif or Scholte). My sense is that the idea of combining arbitration and trajectory adaption has been successfully shown (also, without interaction issues at the low-level control), but only in driving simulators. In my view, it is the implementation in real vehicles (with the proposed controller, that allows flexible control within limitations of mass-produced vehicles) that forms the essence of the contribution. Perhaps phrasing it like that would constitute a clearer story line, that better motivates why much of the paper revolves around the low-level controller (interaction), and its validation, and much less on the other two elements.

RESULTS

The proposed metrics require more detail. The authors do not describe how DrE and StE are normalised, please do so. Also, the StE equation is not given, but referred to in a paper. Please give the equation here, to avoid that the reader has to search for the publication, and give all the information to replicate the calculation.

Note that in my opinion the analysis is rather limited: summation of torque certainly captures something relevant (but how does this normalised summation compare to more standard approaches in the literature like RMS?). I'm not sure about the entropy (how does this compare to metrics like steering reversal rate?). I was actually wondering about other metrics to describe the manoeuvring, like

- the RMS of the lateral difference with respect to the reference trajectory (it seems your multi-level haptic control in Fig13 has smaller difference, compared to the one in Fig 12b

- or consider more safety margin based metrics like time-to-line crossing

CONTRIBUTION AND LIMITATION

I would appreciate a better comparison of the results to the literature. Especially the literature you cited that had two of the three components: how do your results compare to their findings? This would help understand how to read your findings, and further strengthen how your work adds to the literature. Without such links, your results are much less understandable and impactful, and the contribution remains somewhat unclear, in my view.

-----Smaller points-----

I still find "inclusion" a bit of a general term when you actually mean trajectory adaption... It's good that you define it, but do consider just using trajectory adapting.

Figure 2 nicely illustrates "inclusion" and "arbitration", consider also visualising "interaction"

Not sure how EEG would constitute an improvement for your EKF, in terms of extreme changes in stiffness/damping. The relation between EEG and stiffness/damping is not highly correlated.

In the text you call DrE driving effort, in the y-axes driver effort (please harmonise)

Finally, I still see somewhat overstated claims. For example "The superiority of the proposed framework has been validated..." When I examine the data upon which you seem to base this, I only see strong effect size for normalised driver effort (not for steering entropy), and only for modes 3 and 4, compared to 1-2. And even that is probably not statistically significant, given the effect size and sample size, as you rightfully acknowledge at then. Still, if that's the case, I suggest to phrase your claims less absolutely.

The authors would like to thank the reviewers for their thorough work again. We appreciate the constructive comments. Our answers to the reviewers are highlighted in blue.

Reviewer #2:

Thanks for addressing the raised issue. Very nice work.

Reviewer #3:

I am pleased to see the authors took the feedback from myself and the other reviewers seriously, thoroughly revising the introduction, improving the clarity of figures and blockschemes, running additional experiments, and sharpening the conclusions. There is still work to be done, in my opinion, to further clarify the (1) motivation, the (2) value with respect to other approaches in literature, and (3) some of the analyses. I propose another revision, to address these comments.

INTRODUCTION

The introduction has been greatly improved, is much more legible and to the point. Having said that, I still have some concerns.

A small concern lies around the claims of what constitutes shared control. For example, "Shared steering control is available under a preset driver torque threshold". This is not necessarily the case, there are also many examples where shared control is never switched off like that (actually, with sufficiently high torque tuning and such switches, shared control becomes traded control). I would advise clearly defining shared control, and relating your definitions to those proposed in a relevant review paper of shared control, that proposes a clear definition around shared control:

This sentence is not intended as a definition of shared control; instead, it explains how shared control is applied to the development of conventional LCA. However, to clarify this point, the manuscript has been revised by adding the definition of shared control

based on the literature the reviewer listed.

---Abbink, D. A., Carlson, T., Mulder, M., De Winter, J. C., Aminravan, F., Gibo, T. L., & Boer, E. R. (2018). A topology of shared control systems—finding common ground in diversity. *IEEE Transactions on Human-Machine Systems*, 48(5), 509-525.

"Nature of the issue". I like this heading, it provides a clear anchor to the reader. However, when I read the section, I still was unclear about the real nature of the issue.

Why is there a need for the "development of a generic control framework consistent across all cognitive levels"? What cognitive levels do you mean, and what is the trouble of a framework that is "inconsistent" across these? Perhaps you hint here at what you later state more clearly (The integration of interaction, arbitration, and inclusion into a generic multi-level control framework represents the main contribution of this work.) If that is the case, please clarify this accordingly, or else clarify what you mean with "cognitive levels".

To clarify the need for the proposed multi-level shared control framework, the issues related to conventional ADAS issues are recapitulated in more detail in "Nature of the Issue". In addition, it has been highlighted that "Partially automated vehicles are characterized by limited functional integration and discontinuous operation of the ADAS". Further, "cognitive levels" has been replaced by "both tactical and operational vehicle control levels as well as across all levels of automation", following the reviewer's suggestion.

Second, the "major disadvantages" of what the author calls blended control are not completely clarified yet. These issues seem to revolve around

- lowered tracking performance, due to tuning of a PID controller (perhaps adding the PID controller in Fig 1 or 2 would help clarify the control-theoretic issue you address)
- driver comfort and acceptance
- a trade-off (as mentioned in the conclusions also), but without clarifying that trade-off, or the consequences of that trade-off

The trade-off between tracking performance and driver intervention performance has been explained in our paper [13]. In blended control, in case the angle control is designed to accurately track the AD target angle, the driver intervention is rejected as a disturbance. To avoid this, modulation of angle controller gain is required to accept the

driver intervention, which causes the lower angle tracking performance. This is solved if modulation of the angle controller is executed based on the driver activity (hand-off: high angle tracking gain, hands-on: low angle tracking gain), however, this is technically challenging because of the low reliability of the available sensor.

Further, blended control, where haptics change due to disturbances, increases the driver workload and leads to distrust of the automation. Admittance control that can achieve high-fidelity haptics is appropriate for safe and comfortable automation.

Central seems to be: "Ideally, perfect tracking is expected in the absence of driver input, while low rejection performance is required to enable manual intervention." But, there are examples of control architectures that do not suffer from this. For example

- the works of Benloucif et al that you cite,
- or architectures described in papers you don't cite (like van Paassen et al; Scholtens et al; see refs below) - that also include what you call "inclusion"

---Paassen et al. (2017). Four design choices for haptic shared control. In *Advances in Aviation Psychology*, Volume 2 (pp. 237-254)

---Scholtens, W. et al. (2018, October). A new haptic shared controller reducing steering conflicts. In *2018 IEEE International Conference on Systems, Man, and Cybernetics (SMC)* (pp. 2705-2710). IEEE.

- similarly, you currently acknowledge only a limited cross-section of studies that do trajectory adaptation. There are more studies that can both generate haptic control in real-vehicles as well as adapt trajectories (e.g. Kolekar et al, see ref below).

---Kolekar, S., Petermeijer, B., Boer, E., de Winter, J., & Abbink, D. (2021). A risk field-based metric correlates with driver's perceived risk in manual and automated driving: A test-track study. *Transportation research part C: emerging technologies*, 133, 103428.

There are also other ways to do haptic shared control that can adapt trajectories when (double) lane changing, that do not require arbitration. how does your approach compare to that? See:

---Tsoi, K. K., Mulder, M., & Abbink, D. A. (2010). Balancing safety and support: Changing lanes with a haptic lane-keeping support system. In *2010 IEEE international conference on systems, man and cybernetics* (pp. 1236-1243). IEEE.

The literature suggested by the reviewer and the one we cited are summarized in the

following table, to highlight the targeted applications, the focus and the issues with the proposed approaches:

Literature	Type	Application	Focus	Issue
Moterl2012	Interaction and Arbitration	Robotics	Role allocation	When the driver stops interacting, the vehicle returns to the original trajectory. Assisted lane change cannot be covered with these approaches.
Li2019			Impedance modification by game theory-based controller	
Takagi2021			Interaction type realization based on the estimated human goal	
Ercan2021		Vehicle	Impedance	
Izadi2020		Steering HILS	modification by MPC	
Benloucif2017	Inclusion only	Driving simulator	Driver torque-based online trajectory adaptation	Interaction and arbitration are omitted. Low-frequency bandwidth of inclusion would make poor steering feel.
Benloucif2019				
Losey2017	Interaction and Inclusion	Robotics	Local deformation of trajectory	Even if inclusion is stopped, the driver can deviate from the original lane thanks to the interaction compliance, which means, however, that competition is not covered.
Losey2019			Online global trajectory modification based on physical interaction	This paper concludes that if the task requires significant interaction with the environment, arbitration is more appropriate than inclusion. This suggests the necessity of arbitration and inclusion properly depending on the situation.
Paassen2017	Interaction + offline trajectory generation	Driving simulator	Multi-layered shared control	Haptic shared control to track the human-like reference trajectory.
Scholtens2018				

				There is no function, which adapts the reference trajectory based on the driver intervention online.
Tsoi2010	Interaction and Inclusion		Switching of LKA and ALC by TLC	Trajectory adaptation way is not discussed. The trajectory is not adapted continuously based on driver intervention. This is merely binary switching of LKA and LCA.
Kolekar2021	Not related	Real vehicle	Risk filed estimation	Not related to arbitration and inclusion

The literature is mainly classified into the following four configurations. Issues in each of these settings are summarized as follows.

1. Interaction + Arbitration

- Moterl2012, Li2019, Takagi2021, Ercan2021, Izadi2020
- Issues: When the driver stops interacting, the vehicle returns to the original trajectory. Hence, this approach cannot cover assisted lane change and trajectory adaptation.

2. Inclusion only

- Benloucif2017, Benloucif2019
- Issues: Low-frequency bandwidth of inclusion would make poor steering feel, and arbitration can provide a richer haptic information exchange in human interactive sensorimotor control.

3. Interaction + inclusion

- Losey2017, Losey2019, Tsoi2010
- Issues: Even if inclusion is stopped, the driver can deviate from the original lane thanks to the interaction compliance, which means competition is not covered. The driver always has priority.

4. Human-like-reference trajectory + Haptic shared control.

- Paassen2017 and Scholtens2018
- Issues: Trajectory adaptation is not included in the online algorithm.

Since the trajectory adaptation in Paassen2017 and Scholtens2018 is not available online as mentioned by the authors (“...HCR was realised by offline model fitting the empirical average driver steering behavior on a driver model...”), it has not been cited in the

manuscript, due to the lack of relevance. Although Tsoi2010 corresponds to the switching of LKA and LCA by time-to-lane-crossing (TLC), it is not a sequential trajectory update that reflects the driver's intentions. It is only relevant in switching LKA or ALC functions in a binary way. The authors mentioned that "...this (judgement by TLC) causes a gradual shift of the reference path from the center of the current lane to center of the future lane..." and how to update the reference trajectory is unclear. Therefore, this is also not cited. Kolekar2021 is not related to arbitration and inclusion if our understanding is correct. The authors agree that the control architecture provided by Benloucif does not certainly suffer from the trade-off between the tracking performance and ease of driver intervention. However, in their approach, the driver interacts with the vehicle dynamics (AD adapted trajectory), not the steering system dynamics. This means the driver may permanently perceive the conflict because the vehicle dynamics is slower than the haptic information exchange bandwidth between the driver and automation interaction. The vehicle dynamics and what the driver perceives from the automation are required to be independently designed. The authors would like to stress that this design independency is not ensured in their approach, and the manuscript has been revised so as to emphasize this point.

All in all, currently the need to combine all of the three elements (interaction/arbitration/inclusion) - and its expected benefit compared to the literature, besides the combination in and of itself - does not follow logically from the introduction still, nor why the Kalman Filtering is necessary (constituting such a large part of the paper). At least I would prefer an even stronger motivation why these three are needed, and what the added value is compared to literature (e.g., the works of Benloucif or Scholte).

Considering the case of humans jointly carrying a desk, there is a step-by-step process in which recognition-matching and negotiation are first conducted at the force level felt in the hands, and the resulting direction of movement (reference trajectory) is determined at the position level. It has been suggested that this negotiation process and synchronization of reference trajectory can vary widely in frequency bandwidth [17][18]. Since the human-machine interaction is assumed to follow the same characteristics as human-human interaction, the authors consider that the proposed multi-level control framework is reasonable. This study proposes a coherent multi-level control framework inspired by the human-human collaboration to realize more natural shared control for the human driver. To strengthen the motivation of this research, the manuscript has

been revised to reinforce the need for the proposed control framework with the following advantages:

1. Compatible with all levels of automation (0 to 4), where the human can still take part in the driving.
2. Integration of the ADAS functions and continuous operation in shared control mode (override free).
3. ADAS functions that satisfy multi-objective requirements related to vehicle motion and driver intent to effectively contribute to better traffic safety.

My sense is that the idea of combining arbitration and trajectory adaption has been successfully shown (also, without interaction issues at the low-level control), but only in driving simulators. In my view, it is the implementation in real vehicles (with the proposed controller, that allows flexible control within limitations of mass-produced vehicles) that forms the essence of the contribution. Perhaps phrasing it like that would constitute a clearer story line, that better motivates why much of the paper revolves around the low-level controller (interaction), and its validation, and much less on the other two elements.

In our opinion, the novelty of this paper is not limited to the implementation of the control framework in real vehicles but also lies in the proposed control framework that enables more active and continuous automation through a combination of arbitration and inclusion. This has been made more explicit in the reviewed manuscript in the “motivation” and the “contribution and limitation” sections.

RESULTS

The proposed metrics require more detail. The authors do not describe how DrE and StE are normalised, please do so.

The normalization process has been added to the manuscript as a footnote. DrE and StE were normalized by min-max normalization method with the maximum value being the mean of mode 1 and the minimum value being zero, the lower bound of the physical value.

Also, the StE equation is not given, but referred to in a paper. Please give the equation here, to avoid that the reader has to search for the publication, and give all the

information to replicate the calculation.

The following calculation process for StE has been added in Appendix A.

Steering entropy calculation process

- Obtain the time-series steering angle data for each sampling time dt (dt = 150ms in this case with reference to the literature).
- The future steering angle is predicted by quadratic Taylor expansion from the past three data points of the steering angle, and the prediction error between the predicted future steering angle and the actual steering angle is obtained.
- Determine the 90%ile value α centered at 0 degrees (α was set to 0.25 from the average value of the participants when driving in manual mode).
- Divide the frequency distribution of the prediction error into nine bins based on the range of α ($-5\alpha, -2.5\alpha, \alpha, -0.5\alpha, 0.5\alpha, \alpha, 2.5\alpha, 5\alpha$).
- Calculate StE from the proportion P_i of each bin using the following formula.

$$StE = \sum_{i=1}^9 P_i \log_9 P_i$$

Note that in my opinion the analysis is rather limited: summation of torque certainly captures something relevant (but how does this normalised summation compare to more standard approaches in the literature like RMS?).

DrE was calculated with reference to the "driving torque steering effort" in prior work [29][53]. Similar trends can be observed in both DrE and RMS of driver torque input, as suggested by the following figure. However, the authors decided to keep the original calculation of DrE consistent with the prior work. The sentence of DrE has been revised slightly, and a calculation formula has been added in Appendix A.

I'm not sure about the entropy (how does this compare to metrics like steering reversal rate?).

The steering entropy and steering reversal rate (SWWR) are similar criteria as they assess the steering angle variation. In fact, they are highly correlated, as described in Fig 4 and 5 of the following literature (except for the white bin in figures).

Markkula, Gustav & Engström, J. (2006). A Steering Wheel Reversal Rate Metric for Assessing Effects of Visual and Cognitive Secondary Task Load.

While the steering entropy is statistically processed for all data, SWWR does not consider small variations below a set threshold. Therefore, the authors would like to keep steering entropy, which allows smoothness to be calculated more accurately. The trends of these two criteria are similar to the present test result. Hence, the same claim is obtained if the steering wheel reversal rate is used. Both data are presented below for reference.

I was actually wondering about other metrics to describe the manoeuvring, like
- the RMS of the lateral difference with respect to the reference trajectory (it seems your multi-level haptic control in Fig13 has smaller difference, compared to the one in Fig 12b

The lateral vehicle error (LtE) confirms the effect of “inclusion”. Unfortunately, it does not capture the variation of arbitration, as shown below. This is because there is the case that the lateral deviation is the same, even when the automation impedance is variable. In particular, as the driver had been asked to operate as smoothly as possible in the four modes in the lane change evaluation, no significant differences in trajectory are found between the modes. From a “haptics” evaluation perspective, the authors consider that an evaluation based on the driver's torque and angle directly felt by the driver would be helpful, more than vehicle motion.

- Lateral error between the actual and adapted trajectory

- or consider more safety margin based metrics like time-to-line crossing

TLC (time-to-line-crossing) is the time it takes to get out of the lane, and the shorter the maneuver, the more dangerous it is. Since this is a vehicle motion-related criterion, it is not appropriate to evaluate the lane change performance for the same reasons given above regarding the lateral error comment.

CONTRIBUTION AND LIMITATION

I would appreciate a better comparison of the results to the literature. Especially the literature you cited that had two of the three components: how do your results compare to their findings? This would help understand how to read your findings, and further strengthen how your work adds to the literature. Without such links, your results are much less understandable and impactful, and the contribution remains somewhat

unclear, in my view.

The “Contribution and limitation” chapter has been amended by adding a “Multi-level haptic control framework” heading at the chapter's beginning to compare the proposed control with prior literature thoroughly. Using the same words “Integration of ADAS and continuous HSC”, “Compatible to all automation levels”, and “multi-objective”, which are advantages of the proposed control framework mentioned in the introduction chapter, the novelty of the proposed control has been made clearer compared to prior work.

-----Smaller points-----

I still find "inclusion" a bit of a general term when you actually mean trajectory adaption... It's good that you define it, but do consider just using trajectory adapting.

As the reviewer pointed out, “inclusion” is generic. However, we would like to keep it because it is easy to use as a one-word expression paired with “Arbitration”. Since the definition of “inclusion” is “the action or state of including or of being included within a group or structure”, the authors consider that “inclusion” is an appropriate expression in this work.

Figure 2 nicely illustrates "inclusion" and "arbitration", consider also visualising "interaction"

“Interaction” has been added instead of “Virtual EPS” since the feedback loop through “Virtual EPS” enables interaction.

Not sure how EEG would constitute an improvement for your EKF, in terms of extreme changes in stiffness/damping. The relation between EEG and stiffness/damping is not highly correlated.

Our impedance estimation process involves estimating the driver goal and using that to estimate the driver impedance. While the EMG sensor is adequate for directly capturing the driver's impedance, the EEG sensor would help improve the driver's target angle estimation accuracy. As a result of the accurate estimation of driver target angle with EEG, the accuracy of driver impedance estimate might be improved. Therefore, the authors would like to keep this sentence as it is.

In the text you call DrE driving effort, in the y-axes driver effort (please harmonise)

The text and labels of figures have been uniformized with “driver effort”.

Finally, I still see somewhat overstated claims. For example "The superiority of the proposed framework has been validated..." When I examine the data upon which you seem to base this, I only see strong effect size for normalised driver effort (not for steering entropy), and only for modes 3 and 4, compared to 1-2. And even that is probably not statistically significant, given the effect size and sample size, as you rightfully acknowledge at then. Still, if that's the case, I suggest to phrase your claims less absolutely.

The related sentence has been deleted.

REVIEWERS' COMMENTS:

Reviewer #3 (Remarks to the Author):

I think the authors have seriously addressed the concerns of all reviewers involved, including my own. I still feel some choices made in phrasing, structure and analysis limit the potential impact of the work, but I think the work is sufficiently advanced to merit publication.